# CQ-DINO: Mitigating Gradient Dilution via Category Queries for Vast Vocabulary Object Detection

**Zhichao Sun** [1,3]     **Huazhang Hu** [2]     **Yidong Ma** [2]     **Gang Liu** [2]     **Yibo Chen** [2]

**Xu Tang** [2]          **Yao Hu** [2]          **Yongchao Xu** [1,3] ✉

[1] **Institute of Artificial Intelligence, School of Computer Science, Wuhan University**

[2] **Xiaohongshu Inc.**          [3] **Hubei Luojia Laboratory**

## Abstract

With the exponential growth of data, traditional object detection methods are increasingly struggling to handle vast vocabulary object detection tasks effectively. We analyze two key limitations of classification-based detectors: **positive gradient dilution**, where rare positive categories receive insufficient learning signals, and **hard negative gradient dilution**, where discriminative gradients are overwhelmed by numerous easy negatives. To address these challenges, we propose **CQ-DINO**, a category query-based object detection framework that reformulates classification as a contrastive task between object queries and learnable category queries. Our method introduces image-guided query selection, which reduces the negative space by adaptively retrieving the top-K relevant categories per image via cross-attention, thereby rebalancing gradient distributions and facilitating implicit hard example mining. Furthermore, CQ-DINO flexibly integrates explicit hierarchical category relationships in structured datasets (*e.g.*, V3Det) or learns implicit category correlations via self-attention in generic datasets (*e.g.*, COCO). Experiments demonstrate that CQ-DINO achieves superior performance on the challenging V3Det benchmark (surpassing previous methods by 2.1% AP) while maintaining competitiveness on COCO. Our work provides a scalable solution for real-world detection systems requiring wide category coverage.

## 1   Introduction

With the rapid expansion of data, developing a robust AI system capable of large-scale object detection has become essential. This necessity is driven by the increasing complexity and diversity of real-world applications, where AI must manage an extensive vocabulary and dynamic environments. Vast vocabularies inherently present hierarchical category structures, as illustrated by classification datasets like ImageNet [8] and Bamboo [49]. Recent detection benchmarks, such as V3Det [40], which feature 13,204 object categories organized in hierarchical structures, highlight the magnitude of this challenge. While object detection has witnessed significant advancements [31, 20, 25, 11], scaling effectively to vast vocabularies remains a substantial challenge.

Category prediction mechanisms can be broadly categorized into four types, as illustrated in Fig. 1. Classification head-based methods employ feed-forward networks (FFNs) with sigmoid activation and

---

✉Corresponding author: Yongchao Xu <yongchao.xu@whu.edu.cn>

*Source Code: https://github.com/FireRedTeam/CQ-DINO

[1]{zhichaosun, yongchao.xu}@whu.edu.cn

[2]{huhuazhang, mayidong, tangshen}@xiaohongshu.com {liugang.spl, nemochen89, yaoohu}@gmail.com

39th Conference on Neural Information Processing Systems (NeurIPS 2025).

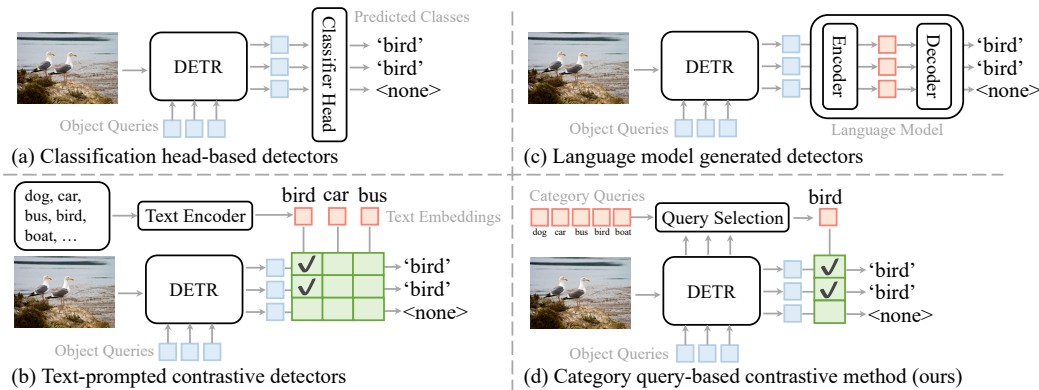

Figure 1: Comparison of category prediction mechanisms for vast vocabulary object detection. (a) Classification head-based detectors with fixed FFN layers face severe optimization challenges with increasing vocabulary size. (b) Text-prompted contrastive detectors leverage VLMs but require multiple inference passes for vast category lists. (c) Language model generated detectors enable open-ended detection but lack control over category granularity. (d) Our proposed CQ-DINO encodes categories as learnable category queries and leverages query selection to identify the most relevant categories in the image, achieving both scalability and improved performance.

Focal Loss [35] optimization. These approaches perform well on benchmarks with limited categories such as COCO [22] (80 categories) and Objects365 [36] (365 categories), but face fundamental challenges and scalability issues in vast vocabulary settings. Text-prompted contrastive methods leverage Vision-Language Models (VLMs) to encode target categories as text inputs, achieving strong open-vocabulary detection [46] performance. However, in vast vocabulary scenarios, text input sequences become prohibitively long, necessitating the splitting of category lists across multiple inference passes. This substantially limits their practical scalability and efficiency. Language model-generated methods approach open-ended [21] object detection by using language models to generate category labels without predefined candidate sets. Although these methods theoretically enable the detection of arbitrary object categories, they generally lack mechanisms to control the granularity of generated labels, which can result in substantial misalignment with practical detection requirements.

In this work, we first systematically analyze the challenges in vast vocabulary detection, focusing particularly on classification-based methods. Our analysis reveals two critical limitations: **(1) positive gradient dilution**, where the sparse positive categories receive insufficient gradient updates compared to the overwhelming negative categories, and **(2) hard negative gradient dilution**, where informative hard negative gradients get overwhelmed among numerous easy negative examples.

To tackle these challenges, we introduce **Category Query-based DINO (CQ-DINO)**, a novel architecture that encodes categories as learnable query embeddings. Our approach centers on image-guided query selection, which identifies relevant categories via category-to-image similarity through cross-attention. The key insight driving our method is that *dynamic sparse category selection significantly reduces the negative search space*. This selection mechanism provides three crucial benefits: (1) balancing the ratio between positive and negative gradients, (2) performing implicit hard mining by selecting the most similar categories, and (3) reducing memory and computational costs, making the framework scalable to extremely large vocabularies. Selected category queries interact with image features to generate object queries. From these object queries, bounding boxes are predicted through a cross-modality decoder. The final classifications are obtained using contrastive alignment between the object queries and the category queries. Unlike traditional classification head-based methods, our category query representation offers greater flexibility by naturally encoding inter-category relationships. For structured datasets with explicit category hierarchies like V3Det [40], we leverage the inherent tree structure to construct hierarchical category queries with an adaptive weighting mechanism that balances local and hierarchical features. For datasets without explicit hierarchies (*e.g.*, COCO [22]), we employ self-attention mechanisms to implicitly learn the correlations between categories.

We evaluate CQ-DINO on both the vast vocabulary V3Det dataset [40] and the standard COCO benchmark [22]. Our approach surpasses previous state-of-the-art methods on V3Det while maintaining competitive results on COCO compared to DETR-based detectors. Our method benefits from vast vocabulary detection while maintaining competitive results in limited vocabulary scenarios.

Our contributions can be summarized as follows:

- We systematically analyze the challenges in vast vocabulary object detection, identifying positive gradient dilution and hard negative gradient dilution as critical limitations of classification-based methods.
- We introduce learnable category queries that flexibly encode category correlations with efficient hierarchical tree construction for explicitly modeling category relationships in vast vocabulary scenarios.
- We develop an image-guided query selection module that dynamically identifies relevant categories per image, effectively addressing the identified limitations while significantly reducing computational complexity.

## 2 Related Work

### 2.1 Vast Vocabulary Object Detection

The progression of object detection benchmarks reflects a steady growth in category vocabulary, evolving from relatively small vocabulary datasets such as PASCAL VOC [10] (20 classes) and COCO [22] (80 classes), to larger vocabulary benchmarks, including Objects365 [36] (365 classes) and Open Images [17] (600 classes). Recently, Wang et al. [40] introduce V3Det, the first vast vocabulary object detection dataset, comprising 13,204 hierarchically structured categories. This unprecedented scale poses significant challenges in terms of scalability and representation.

Recent progress has been driven by the V3Det Challenge [39], yielding several methodological advancements. MixPLv2 proposes a semi-supervised framework that combines labeled V3Det data with unlabeled Objects365 [36] images through pseudo-labeling. RichSem-DINO-FocalNet enhances detection robustness by integrating the RichSem-DINO [28] framework with a FocalNet-Huge backbone [44] pretrained on Objects365. Most recently, Prova [4] introduces multi-modal image-text prototypes specifically optimized for V3Det's fine-grained classification. However, these methods still rely on fundamentally similar classification architectures, raising questions about their effectiveness and scalability for even larger vocabularies beyond tens of thousands of categories.

### 2.2 Object Detectors

**Classification Head-based Methods.** Most object detection frameworks employ feedforward networks (FFNs) as fixed classification heads for category prediction. In two-stage detectors [31, 2, 13], region proposals are generated in a class-agnostic manner by Region Proposal Networks (RPNs), followed by FFN-based region classifiers. By contrast, one-stage detectors [35, 38, 9] predict bounding boxes and categories in a single step. Transformer-based methods, such as DETR [3], reformulate detection as a set prediction problem using learnable queries. Although DETR achieves an elegant end-to-end design, it suffers from slow convergence. Subsequent works [54, 41, 23, 18, 47, 29, 24, 14] mitigate these limitations. For example, Deformable DETR [54] proposes multi-scale deformable attention for sparse spatial sampling. DINO [47] improves performance via contrastive query denoising and mixed query selection. Despite these architectural advances, current methods fundamentally rely on classification heads with activation functions, typically optimized with Focal Loss [35] or Cross-Entropy Loss [50]. This design inherently constrains scalability and presents optimization challenges when extending to vast vocabulary detection scenarios.

**Text-prompted Contrastive Methods.** Vision-language models have advanced the seamless integration of visual and textual modalities for open-vocabulary object detection. These methods encode target categories as text inputs and align visual and textual representations. For instance, GLIP [20] pioneers the use of contrastive learning between image regions and textual phrases. Grounding DINO [25] further improves cross-modal alignment through early fusion of vision and textual features. Similarly, DetCLIP [45] and RegionCLIP [52] leverage image-text pairs with pseudo-labels to enhance region-level semantic understanding and improve generalization. Despite these advances, text-prompted methods face scalability bottlenecks due to the limited capacity of text token inputs during inference. For example, GLIP [20] and Grounding DINO [25] restrict input prompts to approximately 128 tokens per pass, allowing for about 40 categories simultaneously. Thus, detecting all categories in a vast vocabulary benchmark like V3Det [40] (13,204 classes) would require over

*331 sequential inference passes per image.* This constraint renders current text-prompted methods computationally inefficient and impractical for real-time or large-scale detection.

**Language Model-Generated Methods.** Advances in multimodal large language models (MLLMs) have inspired detection methods leveraging their visual understanding and generative abilities. Some MLLMs [5, 1, 37] show preliminary object detection abilities but exhibit limited localization precision and recognition granularity. To mitigate these limitations, recent works [21, 11, 16, 40, 42] use LLMs primarily as category generators rather than direct detectors. For instance, GenerateU [21] employs a T5-based decoder [7] to generate category names from visual features, reframing detection as text generation. Similarly, LLMDet [11] and ChatRex [16] utilize instruction-tuned LLMs to predict object categories from image features. While these generative approaches enable open-ended detection, they often produce inconsistencies due to limited controllability over label granularity. For example, given an image region of a "Persian cat", the model may generate the generic term "cat", causing semantic ambiguity and reduced accuracy for fine-grained detection tasks.

## 2.3 Category Query-based Methods

Learnable queries was popularized in computer vision by DETR [3], marking a paradigm shift from fixed architectural components to task-adaptive representations. Queries serve as learnable embeddings that interact with the visual feature space, enabling the model to capture complex, task-specific patterns. This design has since been adopted in diverse domains, including classification [26], segmentation [6], and multimodal learning [19]. Among these developments, *category queries* represent an innovation introduced by Query2Label [26]. Rather than relying on fixed classification heads, Query2Label proposed learnable category embeddings to capture category-specific features. Subsequent works such as ML-Decoder [34] have outperformed conventional classification methods. The effectiveness of category queries in classification motivated their extension to dense prediction tasks. For example, CQL [43] applies them to human–object interaction classification, ControlCap [51] uses them for semantic guidance in region captioning, and RankSeg [12] integrates them into semantic segmentation, dynamically selecting the top-$k$ most relevant classes during inference. This selective querying reduces the effective search space, improving computational efficiency and segmentation accuracy. While prior works have explored category queries in various contexts, our work addresses a fundamentally different challenge specific to vast vocabulary scenarios. We provide, to the best of our knowledge, the first systematic theoretical analysis of *gradient dilution* issues that arise when dealing with vast category vocabularies, which motivates our image-guided query selection design. Moreover, we introduce a hierarchical tree construction strategy that explicitly models category correlations, enabling effective reasoning over deep semantic hierarchies in vast vocabulary datasets.

## 3 Method

### 3.1 Challenges in Vast Vocabulary Detection

Existing methods for vast vocabulary object detection with $C$ categories ($C > 10^4$), particularly those employing sigmoid-based classifiers with Focal Loss [35], face fundamental optimization challenges. We systematically analyze these issues through a simplified formulation using the Cross-Entropy Loss with sigmoid activation, revealing two critical limitations:

**1) Positive Gradient Dilution.** In vast vocabulary detection, the gradient signal for positive classes is overwhelmed by the aggregated negative gradients. Let $z_c$ denote the logit for class $c$ and $y_c \in \{0, 1\}$ be its ground-truth label. The gradient of the Cross-Entropy Loss $\mathcal{L}$ with respect to $z_c$ is:

$$\nabla_{z_c} \mathcal{L} = \sigma(z_c) - y_c , \qquad (1)$$

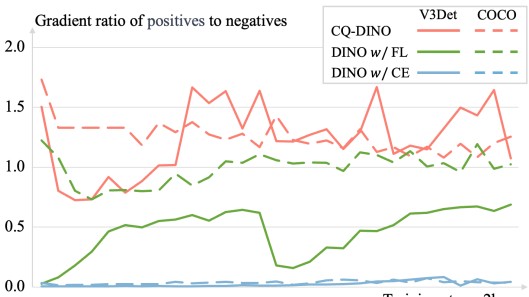

Figure 2: Positive-to-negative gradient ratio comparing CQ-DINO against DINO with Focal Loss (FL) and Cross-Entropy Loss (CE) on V3Det and COCO datasets, showing the initial 2k training iterations where differences are most evident.

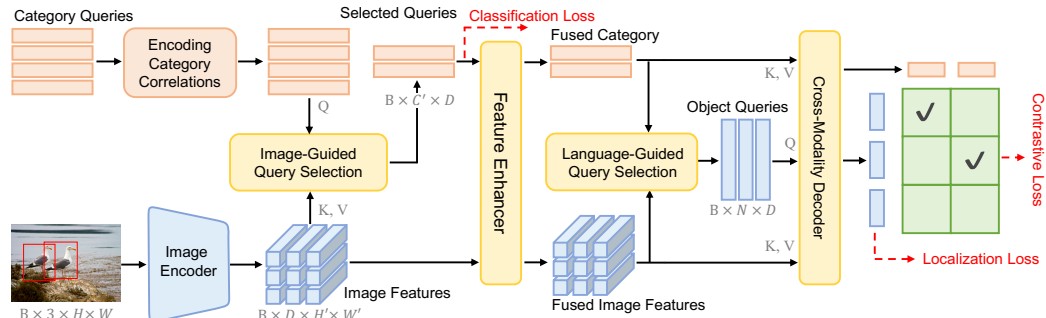

Figure 3: Overview of the CQ-DINO framework for vast vocabulary object detection. Key components: (1) Learnable category queries enhanced with hierarchical tree construction for semantic relationship modeling; (2) Image-guided query selection that identifies the most relevant category queries; (3) Feature enhancer and cross-modality decoder (adapted from GroundingDINO [25]), processing object queries with contrastive alignment between object and selected category queries.

where $\sigma(\cdot)$ is the sigmoid function. For a positive class $c^+$ ($y_{c^+} = 1$), the gradient magnitude is $|\nabla_{z_{c^+}} \mathcal{L}| = 1 - \sigma(z_{c^+})$, while for negatives $c^-$ ($y_{c^-} = 0$), it is $|\nabla_{z_{c^-}} \mathcal{L}| = \sigma(z_{c^-})$.

The total negative gradient magnitude grows linearly with the category count $C$:

$$||\nabla_{z_{c^+}} \mathcal{L}|| \ll \sum_{c^- \neq c^+}^{C} ||\nabla_{z_{c^-}} \mathcal{L}|| \approx (C - 1) \cdot \epsilon, \tag{2}$$

where $\epsilon = \mathbb{E}[\sigma(z_{c^-})]$ represents the average activation probability of negative classes. The positive-to-negative gradient ratio $\rho$ becomes:

$$\rho = \frac{||\nabla_{z_{c^+}} \mathcal{L}||}{\sum_{c^- \neq c^+}^{C} ||\nabla_{z_{c^-}} \mathcal{L}||} \approx \frac{1 - \sigma(z_{c^+})}{(C - 1) \cdot \mathbb{E}[\sigma(z_{c^-})]} \propto \frac{1}{C \cdot \epsilon}. \tag{3}$$

During early training stages, $\epsilon$ retains a non-negligible value. Since $C$ exceeds $10^4$ in vast category scenarios, $\rho \to 0$, causing positive gradients to be suppressed relative to the cumulative negative gradients. This fundamentally hinders the model's ability to learn from positive examples.

**2) Hard Negative Gradient Dilution.** The massive negative space leads to gradient dominance by easily classified negatives rather than informative hard negatives. Let $\mathcal{H}$ denote the set of hard negative classes with $\mathbb{E}[\sigma(z_{c^h})] = \epsilon^h$ for $c^h \in \mathcal{H}$. The ratio of hard negative gradients to total negative gradients is:

$$\eta = \frac{\sum_{c^h \in \mathcal{H}} |\nabla_{z_{c^h}} \mathcal{L}|}{\sum_{c^- \neq c^+}^{C} |\nabla_{z_{c^-}} \mathcal{L}|} \approx \frac{N_h}{C} \cdot \frac{\epsilon^h}{\epsilon}, \tag{4}$$

where $N_h$ is the number of hard negatives. As $C$ exceeds $10^4$, $\eta \to 0$ due to the $\frac{1}{C}$ term, making hard negatives diluted in gradient updates.

Fig. 2 demonstrates these theoretical challenges. The gradient ratio for the V3Det dataset (13,204 classes) is lower than for the COCO dataset (80 classes), revealing the inherent difficulty in vast vocabulary object detection. While Focal Loss partially mitigates these issues by down-weighting easy negatives, the gradient ratio for V3Det remains around 0.5 compared to approximately 1.0 for COCO, indicating that gradient imbalance persists despite these improvements. We provide a comprehensive experimental analysis of Focal Loss performance and limitations in Sec. 4.4.

## 3.2 CQ-DINO

Our key insight is that dynamically selecting a sparse category subset $S \subset \{1, \ldots, C\}$ simultaneously addresses both gradient dilution challenges through gradient magnitude rebalancing and adaptive hard negative mining. As shown in Fig. 3, CQ-DINO consists of three key components:
1) Learnable category queries with correlation encoding. We initialize learnable category queries

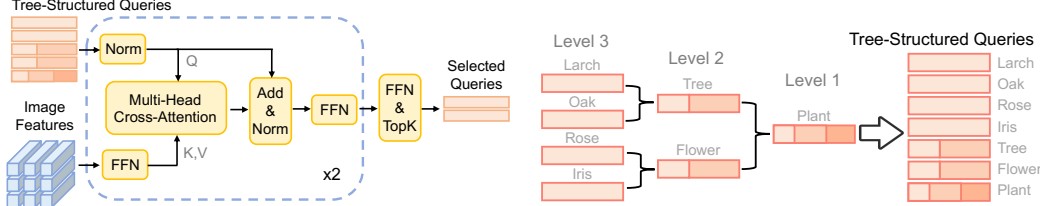

Figure 4: Illustration of image-guided query selection module.

Figure 5: Hierarchical tree construction for category queries.

$Q_{cat} \in \mathbb{R}^{B \times C \times D}$ using the OpenCLIP [30] text encoder, where $B$ is the batch size, $C$ denotes the total number of categories, and $D$ is the embedding dimension. These category queries enable flexible encoding of category correlations through self-attention or hierarchical tree construction (Sec. 3.4).

2) Image-guided query selection. Given image features $F_{img} \in \mathbb{R}^{B \times D \times H' \times W'}$ from the image encoder, we compute similarities between $F_{img}$ and $Q_{cat}$ through multi-head cross-attention modules. For each image, we select the top-$C'$ most relevant queries ($C' \ll C$, typically $C' = 100$ for $C > 10^4$), ensuring that the target class $c^+$ and its most confusing negative classes are preserved. This selection process rebalances the gradients and implicitly performs hard negative mining.

3) Feature enhancer and cross-modality decoder. The selected category queries $Q'_{cat} \in \mathbb{R}^{B \times C' \times D}$ and image features $F_{img}$ are processed through GroundingDINO [25] components. First, the feature enhancer module fuses the category queries and image features. Next, object queries are generated through language-guided query selection. Finally, detection outputs are produced by the cross-modality decoder and contrastive alignment between object queries and selected category queries.

## 3.3 Image-Guided Query Selection

The core innovation of CQ-DINO is our image-guided category selection module, illustrated in Fig. 4. This module employs cross-attention between category queries $Q_{cat}$ and image features $F_{img}$. Here, $Q_{cat}$ serves as queries (Q), while $F_{img}$ provides keys (K) and values (V). The cross-attention layer establishes category-to-image correlations through similarity computation. Then, we apply TopK selection to retain only the top-$C'$ categories ($C' \ll C$) based on activation values. We supervise this selection using Asymmetric Loss [33], which serves as a multi-class classification loss.

The selection rebalances the positive-to-negative gradient ratio. Let $\rho$ and $\rho'$ denote the original and revised positive-to-negative gradient ratios, respectively:

$$\frac{\rho'}{\rho} = \frac{\sum_{c^-}^{C} ||\nabla_{z_{c^-}} \mathcal{L}||}{\sum_{c^- \in S}^{C'} ||\nabla_{z_{c^-}} \mathcal{L}||} \approx \frac{C}{C'}. \tag{5}$$

For a typical setting ($C > 10^4$, $C' = 100$), this achieves a $100\times$ gradient rebalancing factor. Our design provides three benefits: **1) Gradient rebalancing.** By filtering out easy negative categories, our selection module improves the influence of gradients from positive examples. **2) Adaptive hard negative mining.** The selection mechanism ensures that retained negative categories exhibit high semantic relevance to the image content, naturally implementing hard negative mining. **3) Scalable computation.** Processing only $C'$ categories reduces memory consumption and computational cost, making the framework scalable to extremely large vocabularies.

## 3.4 Encoding Category Correlations

A key advantage of our category query approach is the capacity to model complex semantic relationships among categories, which is difficult for traditional classification head-based methods to achieve. We propose two complementary strategies for encoding category correlations.

**Explicit Hierarchical Tree Construction.** For hierarchical datasets like V3Det [40], we introduce hierarchical tree construction, as shown in Fig. 5. The process begins at leaf nodes and progressively integrates hierarchical information upward through the category tree. For each parent node $v$ with children $\mathcal{C}(v)$, its new query $Q'_v$ is a combination of its original query $Q_v$ and the mean pooling of all

Table 1: Comparison with state-of-the-art methods on the V3Det validation set. Best results in each group are highlighted in **bold**.

| Method | Epochs | Backbone | $AP$ | $AP_{50}$ | $AP_{75}$ |
|---|---|---|---|---|---|
| ATSS [48] | 24 | Swin-B | 7.6 | 8.9 | 8.0 |
| FCOS [38] | 24 | Swin-B | 21.0 | 24.8 | 22.3 |
| Faster R-CNN [31] | 24 | Swin-B | 37.6 | 46.0 | 41.1 |
| CenterNet2 [53] | 24 | Swin-B | 39.8 | 46.1 | 42.4 |
| Cascade R-CNN [2] | 24 | Swin-B | 42.5 | 49.1 | 44.9 |
| Deformable DETR [54] | 50 | Swin-B | 42.5 | 48.3 | 44.7 |
| DINO [47] | 24 | Swin-B | 42.0 | 46.8 | 43.9 |
| Prova [4] | 24 | Swin-B | 44.5 | 49.9 | 46.6 |
| CQ-DINO (Ours) | 24 | Swin-B | **46.3** | **51.5** | **48.4** |
| DINO [47] | 24 | Swin-B-22k | 43.4 | 48.4 | 45.4 |
| Prova [4] | 24 | Swin-B-22k | 50.3 | 56.1 | 52.6 |
| CQ-DINO (Ours) | 24 | Swin-B-22k | **52.3** | **57.7** | **54.6** |
| DINO [47] | 24 | Swin-L | 48.5 | 54.3 | 50.7 |
| Prova [4] | 24 | Swin-L | 50.9 | 57.2 | 53.2 |
| CQ-DINO (Ours) | 24 | Swin-L | **53.0** | **58.4** | **55.4** |

direct child nodes. Leaf nodes retain their original features directly, since they have no children.

$$Q'_v = (1 - \alpha_v) \cdot Q_v + \alpha_v \cdot \frac{1}{|\mathcal{C}(v)|} \sum_{c \in \mathcal{C}(v)} Q_c, \tag{6}$$

where $\alpha_v \in [0, 1]$ balances local and hierarchical features.

$$\alpha_v = w \left( 1 + \frac{\log(n_v + 1)}{\log(N_{\max} + 1)} \right), \tag{7}$$

where $n_v$ is the child count for node $v$, $N_{\max}$ is the maximum child count across the tree, and $w \in [0, 0.5]$ is a hyperparameter (default: 0.3). This adaptive weight $\alpha_v$ ensures that parent nodes with more descendants incorporate more collective knowledge, while nodes with fewer children maintain stronger individual semantics.

Building upon this structural design, we introduce a masking strategy during the classification loss computation to mitigate hierarchical ambiguity. If any child category exists in the ground truth, its parent nodes are excluded from the classification loss. This prevents conflicting supervision signals for semantically related categories, such as suppressing "vehicle" when "car" is annotated.

**Implicit Relation Learning.** For categories without an explicit hierarchical structure, we employ a self-attention mechanism to learn category relationships. This allows semantically related categories to influence each other's representations based on learned attention patterns.

## 4 Experiments

### 4.1 Datasets and Implementation Details

We conduct experiments on two detection benchmarks: (1) V3Det [40]: a vast vocabulary detection dataset containing 13,204 categories, with 183k training and 30k validation images. This is our primary benchmark for evaluating vast vocabulary detection. (2) COCO `val2017` [22]: a standard benchmark dataset with 80 object categories, comprising 118k training and 5k validation images. We include this dataset to verify the effectiveness of our method in limited vocabulary scenarios.

Experiments are conducted on 8 A100-40G GPUs with a total batch size of 16, unless otherwise specified. Baseline configurations are used for fair comparison. Three Swin Transformer [27] variants serve as backbones: Swin-B (ImageNet-1k [8] pre-trained), Swin-B-22k (ImageNet-22k pre-trained), and Swin-L (ImageNet-22k pre-trained). Category queries are initialized using CLIP-ViT-L [30] text embeddings. To match dataset vocabularies, we employ 100 category queries for V3Det and 30 for COCO, aligned with their category sizes. The hierarchical tree structure is derived from V3Det's category taxonomy. For implicit relation learning, we use an 8-head self-attention module.

Table 2: Comparison between the proposed CQ-DINO and state-of-the-art DETR variants on COCO val2017, reporting the best results as provided by their respective original papers.

| Method | Epochs | Backbone | $AP$ | $AP_S$ | $AP_M$ | $AP_L$ |
|---|---|---|---|---|---|---|
| H-Def-DETR [15] | 36 | Swin-L | 57.1 | 39.7 | 61.4 | 73.4 |
| Relation-DETR [14] | 12 | Swin-L | 57.8 | 41.2 | 62.1 | **74.4** |
| DINO [47] | 36 | Swin-L | 58.0 | 41.3 | 62.1 | 73.6 |
| Rank-DETR [29] | 36 | Swin-L | 58.2 | 42.4 | **62.2** | 73.6 |
| CQ-DINO (ours) | 24 | Swin-L | **58.5** | **42.5** | 62.1 | 74.0 |

Table 3: Performance of Open-world methods on V3Det dataset. We report zero-shot performance using their strongest models. ** indicates finetuned results.

| Method | Backbone | $AP$ | $AP_{50}$ | $AP_{75}$ |
|---|---|---|---|---|
| GenerateU [21] | Swin-L&T5-B | 0.4 | 0.5 | 0.4 |
| ChatRex [16] | Swin-L&LLM-7B | 1.3 | 1.5 | 1.4 |
| *GenerateU* [21] | Swin-L&T5-B | 21.8 | 27.2 | 22.1 |
| DINO [47] | Swin-L | 48.5 | 54.4 | 50.7 |
| CQ-DINO (Ours) | Swin-L | **53.0** | **58.4** | **55.4** |

Table 4: Ablation study on the effectiveness of encoding category correlations and image-guided query selection components in CQ-DINO on V3Det dataset with Swin-Base-22k backbone. "–" denotes unavailable $AR^C$ metrics due to absence of query selection.

| Encoding Category Correlations | Image-guided Query Selection | $AP$ | $AR^C$ | FPS |
|---|---|---|---|---|
| | | 47.3 | – | 0.7 |
| Hierarchical Tree construction | | 49.4 (↑ 2.1) | – | 0.6 (↓ 0.1) |
| | ✓ | 51.1 | 80.9 | **10.8** |
| Self-Attention module | ✓ | 51.3 (↑ 0.2) | 75.5 (↓ 5.4) | 10.4 (↓ 0.4) |
| Hierarchical Tree Construction | ✓ | **52.3** (↑ 1.2) | 83.3 (↑ 2.4) | 10.6 (↓ 0.2) |

The training objective combines multiple loss terms with the following weights: classification loss (Asymmetric Loss [33], weight=1.0), contrastive alignment (Focal Loss [35], weight=1.0), bounding box regression (L1 Loss, weight=5.0), and GIoU Loss [32] (weight=2.0), as in GroundingDINO [25]. Hungarian matching is used, following GroundingDINO, with identical matching costs for object-to-query assignment. To stabilize training, we use a two-stage approach: first, pre-training category queries, image encoder, and image-guided query selection for 10 epochs to establish high initial target category recall; then, fine-tuning the full detection pipeline in the second stage.

## 4.2 Experimental Results

**Performance on Vast Vocabulary Detection**. Tab. 1 presents our comparison with state-of-the-art methods on the V3Det [40] benchmark. CQ-DINO consistently outperforms all previous approaches across different backbone configurations. With the Swin-B backbone, CQ-DINO achieves 46.3% AP, outperforming general detection methods like Deformable DETR [54] by 3.8% AP and DINO [47] by 4.3% AP. More importantly, CQ-DINO surpasses Prova [4], a specialized vast vocabulary detection method, by 1.8% AP. When integrated with the Swin-B-22k backbone, CQ-DINO achieves 52.3% AP, outperforming Prova by 2.0% AP. With the Swin-L backbone, CQ-DINO achieves 53.0% AP. The consistent improvements across different backbones demonstrate that CQ-DINO effectively addresses vast vocabulary detection challenges.

**Performance on Standard Detection Benchmark**. Tab. 2 compares our method with state-of-the-art approaches on COCO val2017. Despite being primarily designed for vast vocabulary scenarios, CQ-DINO achieves competitive performance, reaching 58.5% AP, which is comparable to recent DETR-based methods. We report the best results from their original papers to ensure a fair comparison. The competitive performance of CQ-DINO is mainly due to the proposed gradient rebalancing and adaptive hard mining strategies.

## 4.3 Ablation Studies

We conduct ablation studies to evaluate the effectiveness of each component in CQ-DINO. Unless otherwise stated, all experiments are performed on the V3Det dataset using a Swin-B-22k backbone.

Table 5: Ablation study on self-attention module (SA) in CQ-DINO on the COCO dataset.

| Method | $AP$ | $AR^C$ | Params (M) |
|---|---|---|---|
| CQ-DINO *w/o* SA | 58.3 | 98.2 | **244.3** |
| CQ-DINO *w/* SA | **58.5** (↑ 0.2) | **99.1** (↑ 0.9) | 246.7 (+2.4) |

Table 6: Ablation study on adaptive weighting in tree construction.

| Method | $AP$ | $AP_{50}$ | $AP_{75}$ | $AR^C$ |
|---|---|---|---|---|
| Fixed weight (0.5) | 51.9 | 57.4 | 54.3 | 82.3 |
| Ours ($\alpha_v$) | **52.3** | **57.7** | **55.4** | **83.3** |

Table 7: Scalability comparison of CQ-DINO with DINO on A100 40G GPU using Swin-B-22k backbone, showing per-category parameters (K), CUDA memory consumption (kB), and maximum supported category capacity (k).

| Method | Params/Cat. (K) | Memory/Cat. (kB) | Max Cats. (k) |
|---|---|---|---|
| DINO [47] | 2.1 | 8.9 | 100 |
| CQ-DINO | 0.8 | 2.7 | 130 |

Table 8: Focal Loss parameter analysis in DINO using Swin-B-22k backbone. AP scores (%) compare different $\alpha$ and $\gamma$ combinations. Dashes "–" indicate unstable training configurations.

| $\gamma$ \ $\alpha$ | 0.25 | 0.35 | 0.50 | 0.75 |
|---|---|---|---|---|
| 2 | 43.4 | 45.1 | **47.4** | – |
| 3 | 43.7 | 43.9 | 45.1 | – |
| 5 | – | – | – | – |

**Effect of Each Component in CQ-DINO.** Table 4 presents the contribution of each component in terms of average precision ($AP$) and category-level average recall with selected queries ($AR^C$). To establish a baseline without image-guided query selection, we conduct experiments on 8 H800-80G GPUs due to memory constraints. Notably, incorporating image-guided query selection increases FPS from 0.7 to 10.8, highlighting its effectiveness in alleviating memory bottlenecks and substantially enhancing inference efficiency. Furthermore, this component addresses the issue of gradient dilution and leads to a substantial AP improvement, from 47.3% to 51.1% in detection performance.

Explicit hierarchical modeling via tree construction leads to an improvement of 1.2% AP and 2.4% $AR^C$. In contrast, employing self-attention for implicit relationship modeling achieves a marginal increase of 0.2% AP but reduces $AR^C$. This is due to difficulties in learning complex relationships across 13k+ categories. Notably, the tree construction method introduces zero additional parameters with only a 0.2 FPS overhead, while the self-attention approach adds 2.36M extra parameters and a 0.4 FPS reduction in inference speed.

While Tab. 4 shows that explicit tree construction outperforms self-attention on V3Det's vast category space, we conduct further experiments on COCO (Table 5). The results reveal that self-attention contributes meaningful 0.2% AP and 0.9% $AR^C$ improvements on datasets with fewer categories. Both experiments validate the effectiveness of encoding category correlations, with the optimal approach depending on the scale of the category space.

**Effectiveness of Adaptive Weighting in Tree Construction.** Tab. 6 shows the importance of our adaptive weighting strategy compared to a fixed weight of 0.5. This approach adjusts weights based on the varying number of child categories for each parent node in the hierarchy. Results show that our adaptive approach outperforms fixed weighting.

### 4.4 Discussion

**Scalability of CQ-DINO.** Tab. 7 compares the scaling efficiency of CQ-DINO and DINO for vast vocabulary detection. CQ-DINO requires only 0.8K parameters per category, representing a 62% reduction from DINO's 2.1K parameters. For runtime memory consumption, CQ-DINO uses 2.7KB CUDA memory per category. To evaluate practical scalability limits, we test the maximum category support on an A100-40G GPU using the Swin-B-22k backbone with a single $800 \times 1333$ resolution input image. CQ-DINO supports detection of up to 130k categories, surpassing DINO's 100k limit. These experiments validate that CQ-DINO enables applications with extremely large vocabularies.

**Limitations of Generation-based Methods.** We evaluate generation-based methods on the V3Det dataset in Tab. 3. Following the evaluation protocol from GenerateU [21], we compute semantic similarity between generated category embeddings and V3Det category embeddings, selecting the highest similarity match as the final prediction. Our experiments reveal poor performance:

GenerateU [21] achieves only 0.4% AP, while the more recent ChatRex [16] achieves just 1.3% AP. This highlights a fundamental limitation: ***generation-based methods struggle to control the granularity of generated categories, creating semantic misalignments with the specific requirements of detection tasks.*** Furthermore, even when we finetune GenerateU on V3Det data, the resulting performance (21.8% AP) still exhibits a substantial gap compared to classification-head methods.

**Focal Loss Parameters Analysis under Gradient Dilution.** As discussed in Sec. 3.1, vast vocabulary detection suffers from gradient dilution challenges. Focal Loss (FL) [35] mitigates this via adaptive weighting with hyperparameters $\alpha$ and $\gamma$. Theoretically, $\alpha$ balances positive/negative sample contributions, and increasing $\alpha$ enhances the model's ability to handle more categories. The factor $\gamma$ focuses learning on hard negatives, where increasing $\gamma$ improves the mining of hard examples. Tab. 8 evaluates FL configurations on DINO. The default setting [40] ($\alpha = 0.25$, $\gamma = 2$) achieves 43.4% AP. Tuning these hyperparameters reveals that training is unstable when $\gamma \geq 5$ or $\alpha \geq 0.75$. Notably, the optimal configuration ($\gamma = 2$ and $\alpha = 0.5$) achieves 47.4% AP, surpassing the baseline by 4.0%. Nonetheless, this is still lower than the 52.3% AP achieved by our CQ-DINO, suggesting that FL hyperparameter tuning, while beneficial, leaves room for further improvement. Interestingly, when examining parameter transferability across architectures in Appendix Tab. 9, we find that this optimal setting does not generalize well. Applying the Swin-B-22k optimal parameters ($\alpha = 0.5$, $\gamma = 2$) to Swin-B degrades performance by 3.3 % AP relative to its default setting. However, they increase the performance by 4.0% AP and 1.6% AP for Swin-B-22k and Swin-L backbones, respectively. These findings suggest that while Focal Loss is effective in addressing the gradient dilution challenge, optimal hyperparameter selection remains architecture-dependent and requires careful tuning.

## 4.5 Limitation

Although CQ-DINO improves vast vocabulary object detection, several limitations remain. First, detection performance is influenced by the recall of the category query selection. Fortunately, CQ-DINO achieves 83.3% $AR^C$. Appendix Tab. 11 shows that increasing the number of category queries improves $AR^C$ but does not lead to higher $AP$. This indicates, in most cases, 83.3% $AR^C$ is not a primary bottleneck. Future work will explore more sophisticated selection strategies to address this gap. Second, the two-stage training paradigm, while efficient in practice (first stage requires only $\sim 1$ hour), may yield suboptimal coordination between stages compared to end-to-end alternatives.

## 5 Conclusion

In this work, we systematically analyze the challenges inherent in vast vocabulary detection: positive gradient dilution and hard negative gradient dilution. Through comprehensive experiments, we expose the limitations of Focal Loss under these challenging settings. To mitigate these issues, we propose CQ-DINO, a novel framework with two core innovations: (1) learnable category queries that encode category correlations, and (2) image-guided query selection that effectively reduces the negative space while performing adaptive hard negative mining. Extensive evaluations on the V3Det and COCO benchmarks demonstrate that CQ-DINO achieves superior performance and strong scalability as vocabulary sizes increase. As future work, we plan to investigate the adaptability of our category query formulation for open-vocabulary detection and incremental learning scenarios.

## Acknowledgments

This work was supported in part by NSFC under Grant 62222112 and 62176186; and in part by the Innovative Research Group Project of Hubei Province under Grant 2024AFA017.

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

# A Appendix

## A.1 Extended Analysis on Gradient Dilution in Vast Vocabulary vs. Class-Imbalanced

In Sec. 3.1 of the main paper, we introduced the concept of positive gradient dilution as a primary challenge in vast vocabulary object detection, using a simplified model assuming a relatively balanced data distribution. This appendix provides a more comprehensive analysis to further clarify the distinction between the gradient dilution problem caused by vast vocabulary size and the one addressed in traditional class-imbalanced learning.

The simplified gradient ratio in our main paper (Eq. (3)) demonstrates that $\rho \propto \frac{1}{C}$, highlighting the direct impact of the vocabulary size $C$. To better distinguish the effects of class imbalance and vocabulary size, we can formulate a more general gradient signal ratio, $\rho_{c+}$, for a positive class $c^+$:

$$\rho_{c+} = \frac{n_{c+} \cdot \|\nabla_{z_{c+}} \mathcal{L}\|}{\sum_{c^- \neq c^+}^{C} n_{c-} \cdot \|\nabla_{z_{c-}} \mathcal{L}\|} \approx \frac{n_{c+} \cdot (1 - \sigma(z_{c+}))}{\mathbb{E}_{c-} [\sigma(z_{c-})] \cdot \sum_{c^- \neq c^+}^{C} n_{c-}}$$

By letting $\epsilon^+$ and $\epsilon^-$ represent the average gradient magnitudes for positive and negative samples respectively, and $N$ be the total sample count across all classes, the ratio can be expressed as:

$$\rho_{c+} \propto \frac{n_{c+} \cdot \epsilon^+}{(N - n_{c+}) \cdot \epsilon^-} \tag{8}$$

This generalized formula reveals two distinct and compounding sources of gradient dilution:

**Class Imbalance:** This well-studied issue is primarily reflected by the term $n_{c+}$ in the numerator. When a class is rare (i.e., has a long-tailed distribution), its sample count $n_{c+}$ is small, which directly reduces the gradient signal ratio $\rho_{c+}$. This is the central challenge that traditional class-imbalance methods aim to solve.

**Vast Vocabulary Size:** This is primarily driven by the term $N - n_{c+}$ in the denominator. In vast vocabulary settings ($C > 10,000$), $N - n_{c+}$ becomes enormous because it aggregates all negative samples from the other $C - 1$ categories. The large $C$ further reduces the ratio $\rho_{c+}$, making the gradient dilution more severe in vast vocabulary object detection. This impacts **both rare and common classes**.

This distinction in the problem's source explains why conventional methods for class imbalance are not sufficient for the vast vocabulary challenge.

## A.2 Algorithm Details

This section provides the detailed algorithmic implementations of the three core components in CQ-DINO, as described in Sec. 3. Each algorithm addresses specific challenges in vast vocabulary object detection:

**Algorithm 1 - Image-Guided Query Selection (Sec. 3.3):** This algorithm implements the core innovation of CQ-DINO by dynamically selecting the most relevant category queries for each image. Through cross-attention mechanisms between category queries and image features, it reduces the negative search space from the full vocabulary to a manageable subset ($C' \ll C$), addressing both positive gradient dilution and hard negative gradient dilution issues identified in Sec. 3.1.

---

**Algorithm 1:** Image-Guided Query Selection

---

**Input:**

$B$: Batch size, $C$: Number of category queries,

$D$: Embedding dimension, $C'$: Number of selected category queries,

$H'$: Height of image features, $W'$: Width of image features,

$\mathbf{Q}_{cat} \in \mathbb{R}^{B \times C \times D}$: Category queries,

$\mathbf{F}_{img} \in \mathbb{R}^{B \times D \times H' \times W'}$: Image features

**Output:**

$\mathbf{Q}'_{cat} \in \mathbb{R}^{B \times C' \times D}$: Selected enhanced category queries,

$\mathcal{I} \in \mathbb{R}^{B \times C'}$: Selection indices

---

```
// Reshape image features for cross-attention
```
$\mathbf{F}_{flat} \leftarrow \text{Reshape}(\mathbf{F}_{img}, [B, D, H' \times W'])$;

$\mathbf{Q}_{enhanced} \leftarrow \mathbf{Q}_{cat}$;

```
// Enhancement through cross-attention layers
```
**for** $l = 1$ **to** $2$ **do**

    ```// Cross-attention with image features```

    $\mathbf{Q}_{attn} \leftarrow \text{MultiHeadCrossAttention}(\mathbf{Q}_{enhanced}, \mathbf{F}_{flat}, \mathbf{F}_{flat})$;

    $\mathbf{Q}_{enhanced} \leftarrow \text{LayerNorm}(\mathbf{Q}_{attn} + \mathbf{Q}_{enhanced})$;

    ```// Feed-forward transformation```

    $\mathbf{Q}_{ffn} \leftarrow \text{FFN}(\mathbf{Q}_{enhanced})$;

    $\mathbf{Q}_{enhanced} \leftarrow \text{LayerNorm}(\mathbf{Q}_{ffn} + \mathbf{Q}_{enhanced})$;

```
// Query selection based on enhanced representations
```
$\mathbf{L} \leftarrow \text{LinearProjection}(\mathbf{Q}_{enhanced})$ // $\mathbf{L} \in \mathbb{R}^{B \times C}$

**for** $b = 1$ **to** $B$ **do**

    $\mathcal{I}_{b,:} \leftarrow \text{TopK}(\mathbf{L}_{b,:}, C')$;

    $\mathbf{Q}'_{cat}[b, :, :] \leftarrow \mathbf{Q}_{enhanced}[b, \mathcal{I}_{b,:}, :]$;

**return** $\mathbf{Q}'_{cat}, \mathcal{I}$

---

**Algorithm 2 - Self-Attention for Implicit Category Relations (Sec. 3.4):** For datasets without explicit hierarchical structures, this algorithm employs multi-head self-attention to learn implicit correlations between categories. It allows semantically related categories to influence each other's representations based on learned attention patterns, complementing the explicit hierarchical approach.

---

**Algorithm 2:** Self-Attention for Implicit Category Relations

---

**Input:** $\mathbf{Q}_{cat} \in \mathbb{R}^{C \times D}$: Category queries,

$H$: Number of attention heads

**Output:** $\mathbf{Q}_{corr} \in \mathbb{R}^{C \times D}$: Correlation-enhanced queries

---

```
// Enhance queries through self-attention mechanism
```
$\mathbf{Q}_{attn} \leftarrow \text{MultiHeadSelfAttention}(\mathbf{Q}_{cat})$;

$\mathbf{Q}_{corr} \leftarrow \text{LayerNorm}(\mathbf{Q}_{attn} + \mathbf{Q}_{cat})$;

**return** $\mathbf{Q}_{corr}$

---

**Algorithm 3 - Explicit Hierarchical Tree Construction (Sec. 3.4):** For datasets with explicit hierarchical structures like V3Det, this algorithm leverages the category hierarchical structures to enhance query representations. It performs bottom-up tree traversal to incorporate hierarchical relationships through adaptive weighting, enabling parent categories to aggregate semantic information from their children while maintaining individual semantics.

**Algorithm 3:** Explicit Hierarchical Tree Construction

---

**Input:** $\mathbf{Q}_{cat} \in \mathbb{R}^{C \times D}$: Category queries,
$\mathcal{T}$: Hierarchical tree structure with nodes $\mathcal{V}$,
$w$: Base weight parameter (default: 0.3)
**Output:** $\mathbf{Q}_{corr} \in \mathbb{R}^{C \times D}$: Correlation-enhanced queries

---

```
// Initialize and prepare tree traversal
```
$\mathbf{Q}_{corr} \leftarrow \mathbf{Q}_{cat}$;
$N_{max} \leftarrow \max_{v \in \mathcal{V}} |\text{Children}(v, \mathcal{T})|$;
$\mathcal{L} \leftarrow \text{TopologicalSort}(\mathcal{T})$;
```
// Bottom-up tree traversal for correlation enhancement
```
**foreach** *node $v$ in Reverse*$(\mathcal{L})$ **do**
    **if** *IsLeaf*$(v, \mathcal{T})$ **then**
```
        // Leaf nodes retain original queries
```
        **continue**;
    **else**
```
        // Compute adaptive weight based on children count
```
        $\mathcal{C}(v) \leftarrow \text{GetChildren}(v, \mathcal{T})$;
        $n_v \leftarrow |\mathcal{C}(v)|$;
        $\alpha_v \leftarrow w + \frac{\log(n_v+1)}{\log(N_{max}+1)}$;
        $\alpha_v \leftarrow \min(\alpha_v, 1.0)$;
```
        // Update parent query with weighted combination
```
        $\mathbf{Q}_{child}^{mean} \leftarrow \frac{1}{n_v} \sum_{c \in \mathcal{C}(v)} \mathbf{Q}_{corr}[c, :]$;
        $\mathbf{Q}_{corr}[v, :] \leftarrow (1 - \alpha_v) \cdot \mathbf{Q}_{cat}[v, :] + \alpha_v \cdot \mathbf{Q}_{child}^{mean}$;

**return** $\mathbf{Q}_{corr}$

---

## A.3 Performance of DINO with Different Focal Loss Parameters

In Tab. 8, we achieve the optimal Focal Loss parameters at $\alpha = 0.5$ and $\gamma = 2$ with the Swin-B-22k backbone. We conduct parameter setting experiments with different backbone in Tab. 9. We find that this optimal setting does not generalize well. The same parameters ($\alpha = 0.5$, $\gamma = 2$) lead to a performance degradation of 3.3% AP for the Swin-B backbone compared to the default configuration ($\alpha = 0.25$, $\gamma = 2$). However, they improve the performance by 4.0% AP and 1.6% AP for the Swin-B-22k and Swin-L backbones, respectively. These findings show that while Focal loss addresses gradient dilution issues, its optimal configuration requires careful parameter tuning.

Table 9: Performance comparison between standard DINO ($\alpha = 0.25$, $\gamma = 2$), ‡ DINO with modified Focal loss parameters ($\alpha = 0.50$, $\gamma = 2$), and the proposed CQ-DINO.

| Method | Backbone | $AP$ | $AP_{50}$ | $AP_{75}$ |
|--------|----------|------|-----------|-----------|
| DINO | Swin-B | 42.0 | 46.8 | 43.9 |
| ‡ DINO | Swin-B | 38.7 (↓ 3.3) | 43.7 (↓ 3.1) | 40.4 (↓ 3.5) |
| CQ-DINO | Swin-B | **46.3** | **51.5** | **48.4** |
| DINO | Swin-B-22k | 43.4 | 48.4 | 45.4 |
| ‡ DINO | Swin-B-22k | 47.4 (↑ 4.0) | 53.3 (↑ 4.9) | 49.7 (↑ 4.3) |
| CQ-DINO | Swin-B-22k | **52.3** | **57.7** | **54.6** |
| DINO | Swin-L | 48.5 | 54.3 | 50.7 |
| ‡ DINO | Swin-L | 50.1 (↑ 1.6) | 56.3 (↑ 2.0) | 52.4 (↑ 1.7) |
| CQ-DINO | Swin-L | **53.0** | **58.4** | **55.4** |

## A.4 Gradient Norm Visualization During Training

As reported in Tab. 8, CQ-DINO achieves superior performance compared to DINO with optimal Focal Loss parameters ($\alpha = 0.5$, $\gamma = 2$), outperforming the default configuration of DINO ($\alpha = 0.25$,

$\gamma = 2$). To analyze their impact on training gradients, we visualize the gradient norm in Fig. 6 for three configurations: (1) CQ-DINO (red), (2) DINO with $\alpha = 0.5$ (blue), and (3) DINO with $\alpha = 0.25$ (orange), all using the Swin-B-22k backbone. The results reveal that the gradient norm for DINO with $\alpha = 0.25$ (orange line) remains low throughout training, indicating insufficient learning from both positive and hard negative samples. In contrast, DINO with $\alpha = 0.5$ (blue line) initially displays strong gradients, but these are unstable and fluctuate considerably, as reflected in the high variance of the blue points. Meanwhile, CQ-DINO maintains a balanced trajectory, sustaining moderate and stable gradient magnitudes during the entire training process. This sustained and balanced gradient norm demonstrates that CQ-DINO more effectively addresses the gradient dilution issues.

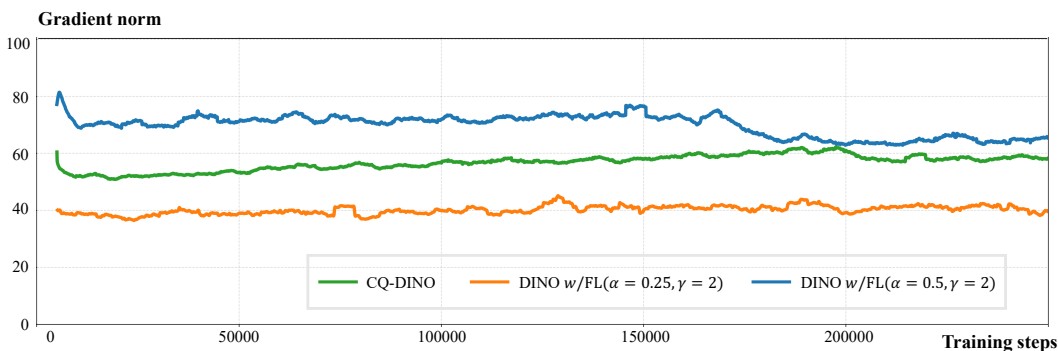

Figure 6: Gradient norm visualization during training process.

## A.5 Extended Gradient Ratio Analysis

**Positive-to-Negative Gradient Ratio.** We extend our analysis to 400k training iterations to examine the effect of CQ-DINO on gradient distribution. As illustrated in Fig. 7, CQ-DINO mitigates the positive gradient dilution problem inherent in vast vocabulary tasks. During the early training stage (fewer than 10k iterations; see Fig. 2), CQ-DINO maintains a substantially higher positive-to-negative gradient ratio compared to DINO with cross-entropy (CE) or focal loss (FL). Moreover, CQ-DINO reaches a balanced state (>1.0) earlier than baselines such as DINO w/CE and DINO w/FL.

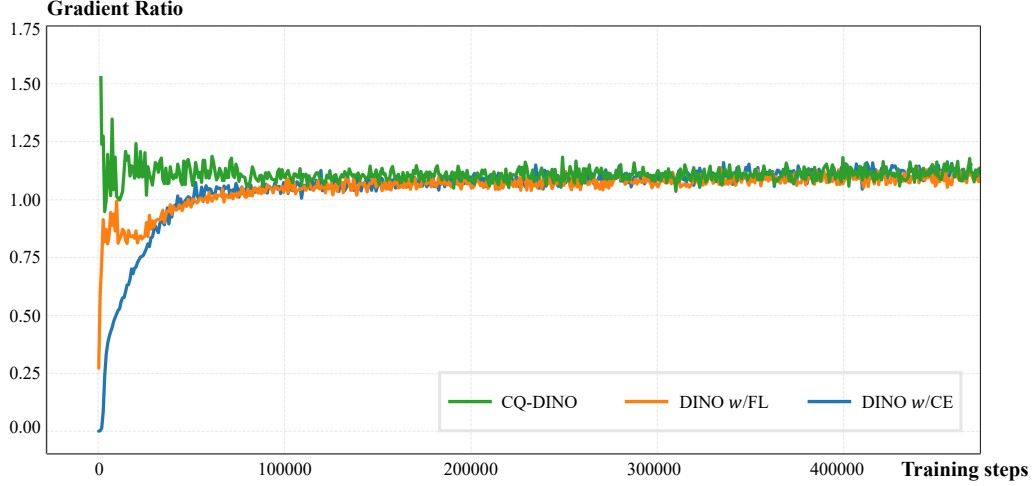

Figure 7: Positive-to-negative gradient ratio across extended training iterations on V3Det dataset.

**Hard-Negative Gradient Contribution.** We define hard negatives as the top 10% of negative categories with the highest prediction scores, representing the most confusing negatives that require

focused learning. Fig. 8 presents the proportion of total negative-gradient magnitude attributable to hard negatives. In early training stage, CQ-DINO achieves higher hard-negative ratios compared to DINO w/CE and DINO w/FL. This confirms that our image-guided query selection implicitly performs effective hard-negative mining by filtering irrelevant categories and concentrating on informative ones.

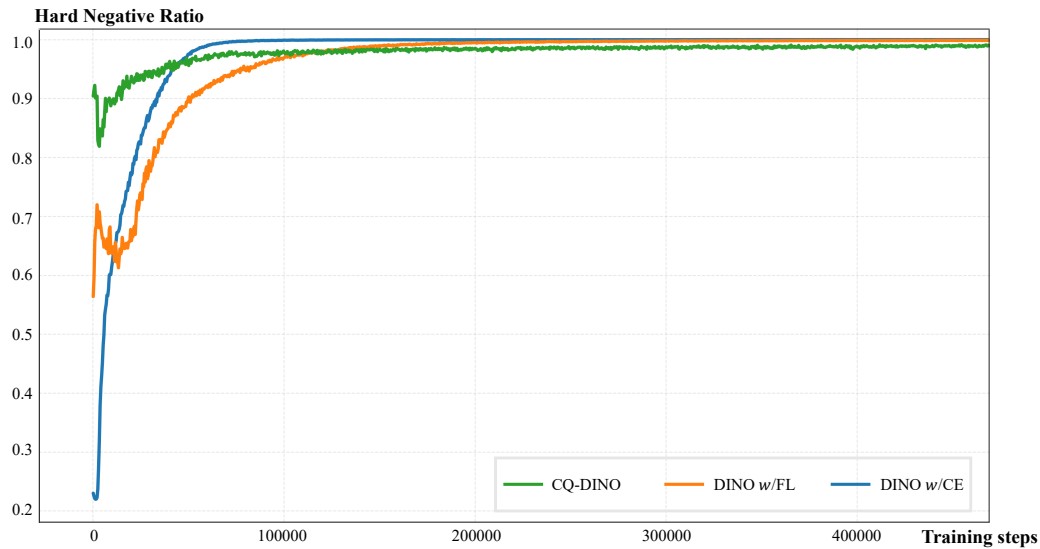

Figure 8: Hard negative-to-all negative gradient ratio across training iterations on V3Det dataset.

## A.6 Additional Ablations on Isolated Component Contributions

To provide a comprehensive understanding of our method's effectiveness and isolate the contributions of individual components, we conduct additional ablation studies that complement the analysis presented in Tab. 4. We analyze the following components, with results presented in Tab. 10:

1. Baseline Architecture: The foundational model based on Grounding DINO, including its feature enhancer and decoder. This corresponds to the setting where all our proposed components are disabled (row 1).

2. Training-stage Gradient Dilution Mitigation: Our proposed image-guided query selection method applied during training. This ensures that the model's text encoder and fusion layers receive focused gradient signals from a small subset of relevant categories for each image.

3. Inference-stage Category Selection: A computational efficiency mechanism that selects only the top-K most relevant categories based on similarity between category queries and image features during inference.

4. Encoding Category Correlations: the hierarchical tree construction method, which was identified as the effective approach for vast vocabulary setting.

The detailed ablations in Tab. 10 confirm three main observations:

**Training-stage gradient dilution mitigation is the primary contributor.** On V3Det, introducing this component yields a +1.9% AP gain (row 2 vs. row 4), validating our hypothesis that gradient dilution poses a significant challenge in large-vocabulary detection. On COCO, where the label space is much smaller, the improvement is modest (+0.3% AP).

**Inference-stage category selection alone does not resolve gradient dilution.** On V3Det, applying inference selection without training-stage mitigation reduces AP (row 2 vs. row 3), likely because the underlying gradients remain diluted during training. In contrast, on COCO, inference selection provides a mild AP improvement due to reduced background competition.

Table 10: Detailed ablation study on the isolated contributions of our proposed components. "0" denotes the absence of the corresponding component, "1" denotes its inclusion.

| Training-stage gradient dilution | Inference-stage category selection | Encoding category correlations | V3Det | | COCO | |
|---|---|---|---|---|---|---|
| | | | AP | FPS | AP | FPS |
| 0 | 0 | 0 | 47.3 | 0.7 | 57.5 | 9.8 |
| 0 | 0 | 1 | 49.4 | 0.6 | 57.9 | 9.7 |
| 0 | 1 | 1 | 46.6 | 10.6 | 58.2 | 9.9 |
| 1 | 0 | 1 | 51.3 | 0.6 | 58.2 | 9.7 |
| 1 | 1 | 0 | 51.1 | 10.8 | 58.3 | 10.0 |
| 1 | 1 | 1 | **52.3** | 10.6 | **58.5** | 9.9 |

**Combination leads to best accuracy–efficiency trade-off.** Integrating both training-stage gradient dilution mitigation and inference-stage category selection (row 6) achieves the highest AP on both datasets while providing substantial speedups: on V3Det, FPS improves from 0.6 to 10.6.

### A.7 Impact of Category Query Count

Tab. 11 presents the impact of varying the number of category queries (50, 100, 200) on performance. As expected, increasing the number of queries leads to higher $AR^C$ scores, at the cost of reduced inference speed. However, simply increasing the query count is not always beneficial. Excessive queries lead to gradient imbalance between positive and negative examples, resulting in diminishing performance gains. Furthermore, improvements in AR do not necessarily correlate with increases in AP, as reflected in our observation that an $AR^C$ of 83.3% is not the limiting factor at the current stage. Empirically, we find that 100 queries achieve an optimal balance between detection performance and computational efficiency for V3Det. On the COCO dataset, 30 queries are sufficient, achieving 99.1% $AR^C$.

Table 11: Ablation study with different numbers of selected category queries in CQ-DINO on V3Det.

| Query Count | $AP$ | $AR^C$ | FPS |
|---|---|---|---|
| 50 | 51.8 | 78.4 | **6.9** |
| 100 | **52.3** | 83.3 | 6.6 |
| 200 | 52.2 | **87.5** | 5.9 |

### A.8 Failure Case Analysis

To better understand the limitations of our approach and identify key bottlenecks, we conduct a systematic failure case analysis of CQ-DINO. Specifically, we examine the 3,527 categories for which the category-level recall falls below 83.3%. This analysis highlights category frequency and object scale as the dominant factors behind performance drops.

**Analysis by Category Frequency.** Among the 3,527 low-recall categories, 374 are rare and 3,153 are common, following our defined frequency threshold. As shown in Tab. 12, CQ-DINO achieves 20.5% AP on rare categories, which is significantly lower than our overall performance of 52.3% AP. This gap underscores the inherent difficulty of rare-category detection in large-vocabulary settings. Notably, despite the challenge, CQ-DINO consistently surpasses both the vanilla DINO and a DINO variant optimized with Focal Loss, indicating that the drop in rare-category performance is a domain-wide challenge rather than a limitation specific to our model design.

**Analysis by Object Scale** We further assess performance across object scales (Tab. 13). The largest gap appears for small objects, where CQ-DINO achieves only 14.1% AP, far below the overall 52.3% AP. While performance on medium- and large-scale objects is robust, this finding highlights that small object detection remains a critical challenge in vast vocabulary settings.

Table 12: Performance on category frequency with category-level recall below 83.3%.

| Method | Rare | Common | All |
|---|---|---|---|
| DINO (Focal $\alpha$=0.50, $\gamma$=2) | 15.5 | 27.1 | 25.8 |
| DINO (Focal $\alpha$=0.50, $\gamma$=2) | 19.9 | 31.1 | 29.9 |
| **CQ-DINO** | **20.5** | **32.1** | **30.9** |

Table 13: Performance across object scales in low-recall cases.

| Method | Small | Middle | Large |
|---|---|---|---|
| DINO | 9.1 | 16.5 | 33.8 |
| DINO (Focal $\alpha$=0.50, $\gamma$=2) | 11.6 | 20.2 | 38.3 |
| **CQ-DINO** | **14.1** | **23.1** | **38.9** |

