# OpenReview forum: "CQ-DINO: Mitigating Gradient Dilution via Category Queries for Vast Vocabulary Object Detection"
_NeurIPS.cc/2025/Conference — NeurIPS 2025 poster_

### Official Review · Reviewer_9iMy · 2025-07-01

**Clarity:** 2
**Significance:** 2
**Originality:** 3
**Rating:** 4
**Confidence:** 4

**Summary:**

In this paper, the authors propose CQ-DINO, a DINO variant for vast-vocabulary object detection. To effectively reduce the effect from huge label space, CQ-DINO leverages Image-Guided Query Selection, which can reduce the label space and rebalance the gradient during training. To build category correlations for precise query selection, the authors propose Hierarchical Tree Construction and Implicit Relation Learning. Experimental show that CQ-DINO effectively improves the performance of vast-vocabulary object detection.

**Questions:**

See the weakness part.

**Ethical Concerns:**

["NO or VERY MINOR ethics concerns only"]

**Final Justification:**

The authors have addressed my concerns regarding unclear description and key comparison.

**Limitations:**

yes.

**Quality:**

3

**Strengths And Weaknesses:**

## Strength

1. The performance is promising.

2. The idea is intuitive and reasonable.

## Weakness

1. The most critical weakness of this paper is the description of Image-Guided Query Selection and Encoding Category Correlations. The technical details are omitted, which limits the expression of both motivation of this method and merits of proposed methods. This drawback restricts the pre-rebuttal score. We recommend the authors to 1) clearly formulate the process of Image-Guided Query Selection by equation of algorithm, 2) compare the proposed selection methods with other solutions (e.g., using off-the-shelf classifiers), 3) clearly formulate the process of Encoding Category Correlations. These improvements can improve the readability of this paper.

2. The gradient norm shown in Fig 6 is not sufficient to show the effectiveness of rebalancing gradient and gradient dilution. The authors should provide additional analysis to explicitly investigate the change in gradient during training.

---

> ### Author Rebuttal · Authors · 2025-07-31
>
> Thank you for the careful reading and kind words. We are pleased that you find our performance promising and our idea intuitive and reasonable. We address your concerns in the following.
> > Q1: Clearly formulate the process of Image-Guided Query Selection by equation of algorithm.
> >
> A1: Thank you for this valuable suggestion. Below, we provide the algorithmic descriptions (which we believe is more clear than equations) of the  Image-Guided Query Selection process. This algorithm share enhance the clarity of our method in the revised paper.
> ### Algorithm 1: Image-Guided Query Selection
> ---
> **Input:**
> - B: Batch size
> - C: Number of category queries
> - D: Embedding dimension
> - C': Number of selected category queries
> - H': Height of image features
> - W': Width of image features
> - $Q_{cat}$: Category queries ∈ $R^{B×C×D}$
> - $F_{img}$: Image features ∈ $R^{B×D×H'×W'}$
>
> **Output:**
> - $Q_{cat}'$: Selected enhanced category queries ∈ $R^{B×C'×D}$
> - $I$: Selection indices ∈ $R^{B×C'}$
> ---
> **1:** $F_{flat}$ ← Reshape($F_{img}$, [B, D, H'×W'])
>
> **2:** $Q_{enhanced}$ ← $Q_{cat}$ &nbsp;&nbsp;&nbsp;&nbsp;*// Initialize enhanced queries*
>
> **3:** **for** l = 1 **to** 2 **do** &nbsp;&nbsp;&nbsp;&nbsp;*// Two cross-attention layers*
>
> **4:** &nbsp;&nbsp;&nbsp;&nbsp;$Q_{attn}$ ← MultiHeadCrossAttention($Q_{enhanced}$, $F_{flat}$, $F_{flat}$) &nbsp;&nbsp;&nbsp;&nbsp;*// Cross-attention*
>
> **5:** &nbsp;&nbsp;&nbsp;&nbsp;$Q_{enhanced}$ ← LayerNorm($Q_{attn}$ + $Q_{enhanced}$) &nbsp;&nbsp;&nbsp;&nbsp;*// Add & Norm*
>
> **6:** &nbsp;&nbsp;&nbsp;&nbsp;$Q_{ffn}$ ← FFN($Q_{enhanced}$) &nbsp;&nbsp;&nbsp;&nbsp;*// Feed-forward network*
>
> **7:** **end for**
>
> **8:** $L$ ← LinearProjection($Q_{enhanced}$) &nbsp;&nbsp;&nbsp;&nbsp;*// Compute selection logits* ∈ $R^{B×C}$
>
> **9:** **for** b = 1 **to** B **do**
>
> **10:** &nbsp;&nbsp;&nbsp;&nbsp;$I_{(b,:)}$ ← TopK($L_{(b,:)}$, C') &nbsp;&nbsp;&nbsp;&nbsp;*// Select top-C' indices*
>
> **11:** &nbsp;&nbsp;&nbsp;&nbsp;$Q_{cat}'$[b,:,:] ← $Q_{enhanced}$[b, $I_{(b,:)}$, :] &nbsp;&nbsp;&nbsp;&nbsp;*// Select enhanced queries*
>
> **12:** **end for**
>
> **13:** **return** $Q_{cat}'$, $I$
>
> ---
> > Q2: Compare the proposed selection methods with other solutions (e.g., using off-the-shelf classifiers)
> >
> A2: Following your suggestion, we conducted experiments comparing our end-to-end image-guided query selection with alternative selection strategies to demonstrate the necessity of joint optimization. We designed two alternative approaches for the query selection mechanism:
>
> 1.Frozen Selection: We pre-trained all encoder components (image encoder, category queries, and image-guided query selection module) for classification, then froze these components entirely during detection training.
>
> 2.Dual-Encoder Off-the-shelf: Based on the frozen selection method, we duplicated the image encoder architecture. One copy remains frozen together with category queries and selection modules to serve as an off-the-shelf classifier for category selection, while the second image encoder continues training for the detection task. This approach represents a compromise between using off-the-shelf components and task-specific optimization, separating classification and detection feature learning.
>
> Table R-9: Comparison with alternative selection strategies.
>
> | Method | AP | $AR^C$ | FPS | Training memory |
> | --- | --- | --- | --- | --- |
> | Frozen selection | 50.5 | 85.7 | 10.6 | 28.2GB |
> | Dual-encoder off-the-shelf | 51.9 | 85.7 | 7.9 | 46.7GB |
> | CQ-DINO (End-to-end) | 52.3 | 83.3 | 10.6 | 33.3GB |
>
> Our end-to-end approach achieves the highest performance, outperforming the frozen selection by 1.8% AP and the dual-encoder approach by 0.4% AP. This demonstrates that joint optimization is important. Though the dual-encoder approach achieves similar results to our CQ-DINO (51.9% vs 52.3% AP), its main weakness lies in the significant computational overhead. It reduces inference speed to from 10.6 to 7.9 FPS while requiring 40% more memory (46.7GB vs 33.3GB). This brings more challenge for real-world deployment despite competitive accuracy.
> > Q3:  Clearly formulate the process of Encoding Category Correlations
> >
> A3: Thank you for this important feedback. We provide a detailed algorithmic formulation of the Category Correlation Encoding process, which will be included in the revised paper.
> ## 2. Encoding Category Correlations
>
> ### Algorithm 2: Explicit Hierarchical Tree Construction
> ---
> **Input:**
> - $Q_{cat}$: Category queries ∈ $R^{C×D}$
> - T: Hierarchical tree structure with nodes V
> - w: Base weight parameter (default: 0.3)
>
> **Output:**
> - $Q_{corr}$: Correlation-enhanced queries ∈ $R^{C×D}$
> ---
> **1:** $Q_{corr}$ ← $Q_{cat}$ &nbsp;&nbsp;&nbsp;&nbsp;*// Initialize with original queries*
>
> **2:** $N_{max}$ ← $max_{v∈V} |Children(v, T)| $&nbsp;&nbsp;&nbsp;&nbsp;*// Find maximum children count*
>
> **3:** L ← TopologicalSort(T) &nbsp;&nbsp;&nbsp;&nbsp;*// Get bottom-up ordering*
>
> **4:** **for each** node v in Reverse(L) **do** &nbsp;&nbsp;&nbsp;&nbsp;*// Bottom-up traversal*
>
> **5:** &nbsp;&nbsp;&nbsp;&nbsp;**if** IsLeaf(v, T) **then**
>
> **6:** &nbsp;&nbsp;&nbsp;&nbsp;&nbsp;&nbsp;&nbsp;**continue** &nbsp;&nbsp;&nbsp;&nbsp;*// Keep original query for leaf nodes*
>
> **7:** &nbsp;&nbsp;&nbsp;&nbsp;**else**
>
> **8:** &nbsp;&nbsp;&nbsp;&nbsp;&nbsp;&nbsp;&nbsp;C(v) ← GetChildren(v, T) &nbsp;&nbsp;&nbsp;&nbsp;*// Get children of node v*
>
> **9:** &nbsp;&nbsp;&nbsp;&nbsp;&nbsp;&nbsp;&nbsp;$n_v$ ← |C(v)| &nbsp;&nbsp;&nbsp;&nbsp;*// Count children*
>
> **10:** &nbsp;&nbsp;&nbsp;&nbsp;&nbsp;&nbsp;&nbsp;$α_v ← w + log(n_v + 1)/log(N_{max} + 1)$ &nbsp;&nbsp;&nbsp;&nbsp;*// Adaptive weight (Eq. 7)*
>
> **11:** &nbsp;&nbsp;&nbsp;&nbsp;&nbsp;&nbsp;&nbsp;$α_v ← min(α_v, 1.0)$ &nbsp;&nbsp;&nbsp;&nbsp;*// Ensure $α_v$ ∈ [0, 1]*
>
> **12:** &nbsp;&nbsp;&nbsp;&nbsp;&nbsp;&nbsp;&nbsp;$Q_{child}^{mean} ← (1/n_v) ∑_{c∈C(v)} Q_{corr}[c,:]$ &nbsp;&nbsp;&nbsp;&nbsp;*// Mean pooling*
>
> **13:** &nbsp;&nbsp;&nbsp;&nbsp;&nbsp;&nbsp;&nbsp;$Q_{corr}[v,:] ← (1-α_v)·Q_{cat}[v,:] + α_v·Q_{child}^{mean}$ &nbsp;&nbsp;&nbsp;*// Update (Eq. 6)*
>
> **14:** &nbsp;&nbsp;&nbsp;&nbsp;**end if**
>
> **15:** **end for**
>
> **16:** **return** $Q_{corr}$
>
> ---
>
> ### Algorithm 3: Self-Attention for Implicit Category Relations
> ---
> **Input:**
> - $Q_{cat}$: Category queries ∈ $R^{C×D}$
> - H: Number of attention heads
>
> **Output:**
> - $Q_{corr}$: Correlation-enhanced queries ∈ $R^{C×D}$
> ---
> **1:** $Q_{attn}$ ← MultiHeadSelfAttention($Q_{cat}$) &nbsp;&nbsp;&nbsp;&nbsp;*// Multi-head self-attention*
>
> **2:** $Q_{corr}$ ← LayerNorm($Q_{attn}$ + $Q_{cat}$) &nbsp;&nbsp;&nbsp;&nbsp;*// Add & Norm*
>
> **3:** **return** $Q_{corr}$
>
> ---
> > Q4. More **explicit gradient change analysis to show the effectiveness of rebalancing gradient and gradient dilution.**
> >
> A4: We provide comprehensive gradient change analysis with extended training iterations to explicitly demonstrate the effectiveness of our gradient rebalancing mechanism in addressing both positive gradient dilution and hard negative gradient dilution.
>
> 1.Extended Positive-to-Negative Gradient Ratio Analysis
>
> We extend the training iterations in Fig 2 as shown in Table R-6. Due to policy constraints that restrict the inclusion of images, we present the extended analysis through sampled key iterations across 200k training steps.
>
> Results show that CQ-DINO maintains early rebalancing advantage. CQ-DINO maintains consistently higher ratios (>1.0) throughout the critical early training phase (less than 10k iterations), when gradient balance is most crucial for learning meaningful representations. In contrast, DINO with Cross-Entropy loss starts with critically low positive gradient ratios (0.01 at 0.5k iterations), demonstrating severe positive gradient dilution.
>
> 2.Hard Negative Gradient Dilution Analysis
>
> To examine the hard negative gradient dilution problem discussed in Section 3.1, we define hard negatives as the top 10% of categories with the highest prediction scores among all negative categories, representing the most confusing negatives requiring focused learning.
>
> Results in Table R-8 show that CQ-DINO consistently achieves the highest hard negative ratios throughout training, particularly during critical early stages (0.83 vs 0.23 for DINO w/CE at 0.5k iterations). The consistently high ratios confirm our theoretical analysis that image-guided query selection performs effective implicit hard negative mining by naturally selecting the most semantically challenging and relevant categories.
>
> The results validate our theoretical analysis and demonstrate the practical effectiveness of image-guided query selection in vast vocabulary detection scenarios. The original plots will be included in the revised paper.
>
> Table R-6:  Positive-to-negative gradient ratio across extended training iterations.
>
> | Iter | 0.5k | 1k | 1.5k | 2k | 2.5k | 3k | 5k | 7k | 10k | 15k | 25k | 50k | 75k | 100k | 200k |
> | --- | --- | --- | --- | --- | --- | --- | --- | --- | --- | --- | --- | --- | --- | --- | --- |
> | DINO w/CE | 0.01 | 0.09 | 0.11 | 0.06 | 0.11 | 0.28 | 0.43 | 0.62 | 0.79 | 0.93 | 1.00 | 1.10 | 1.04 | 1.12 | 1.10 |
> | DINO w/FL | 0.63 | 0.31 | 0.47 | 0.68 | 0.87 | 0.77 | 0.91 | 0.84 | 0.88 | 0.95 | 1.10 | 1.07 | 1.09 | 1.17 | 1.12 |
> | CQ-DINO | 1.08 | 1.56 | 1.21 | 1.64 | 1.70 | 1.39 | 1.50 | 1.23 | 1.12 | 1.18 | 1.12 | 1.19 | 1.16 | 1.08 | 1.14 |
>
> Table R-8: Hard negatives-to-all negative gradient ratio across different training iterations
>
> | Iter | 0.5k | 1k | 1.5k | 2k | 2.5k | 3k | 5k | 7k | 10k | 15k | 25k | 50k | 75k | 100k | 200k |
> | --- | --- | --- | --- | --- | --- | --- | --- | --- | --- | --- | --- | --- | --- | --- | --- |
> | DINO w/CE | 0.23 | 0.53 | 0.61 | 0.67 | 0.72 | 0.77 | 0.89 | 0.97 | 0.99 | 0.99 | 1.00 | 1.00 | 1.00 | 1.00 | 1.00 |
> | DINO w/FL | 0.69 | 0.65 | 0.63 | 0.62 | 0.64 | 0.66 | 0.81 | 0.88 | 0.93 | 0.97 | 0.99 | 0.99 | 0.99 | 0.99 | 0.99 |
> | CQ-DINO | 0.83 | 0.90 | 0.89 | 0.91 | 0.92 | 0.93 | 0.94 | 0.96 | 0.98 | 0.98 | 0.98 | 0.99 | 0.99 | 0.99 | 0.99 |

---

> > ### Comment · Reviewer_9iMy · 2025-08-04
> >
> > Nice rebuttal, the response has addressed most of my concerns.

---

> > > ### Author Response · Authors · 2025-08-05
> > >
> > > We are glad to hear that our responses have addressed your concerns and questions. We sincerely appreciate you taking the time to read our rebuttal and for your positive feedback. We will incorporate all the suggestions and clarifications into the revised version of the paper. Thank you once again for your valuable, constructive feedback and for your consideration.

---

### Official Review · Reviewer_8AUD · 2025-07-01

**Clarity:** 3
**Significance:** 3
**Originality:** 3
**Rating:** 4
**Confidence:** 4

**Summary:**

This paper addresses the critical challenge of scalable object detection in vast vocabulary scenarios, an increasingly important problem in AI. The authors identify two key issues in classification-based methods: positive gradient dilution and hard negative gradient dilution, which hinder performance in large-scale settings. To overcome these, the proposed Category Query-based DINO (CQ-DINO) introduces learnable category queries and an image-guided query selection mechanism. These innovations dynamically reduce the negative search space, balance gradient updates, and improve computational efficiency. The paper demonstrates strong results on the V3Det dataset, achieving state-of-the-art performance, while also delivering competitive results on COCO. The methodology is well-motivated, with a clear focus on scalability and efficiency, and the experimental results support the claims. However, further analysis on generalization to other datasets and a more detailed comparison with alternative architectures would strengthen the work. Overall, the paper makes a meaningful contribution to vast vocabulary object detection.

**Questions:**

Could the proposed CQ-DINO handle category correlations in datasets with incomplete or ambiguous hierarchies?

**Ethical Concerns:**

["NO or VERY MINOR ethics concerns only"]

**Final Justification:**

I thank the authors for their thorough rebuttal, as well as the additional experiment results. This rebuttal has resolved the majority of my concerns. As a result, I decide to keep my rating as borderline accept.

**Limitations:**

Visualization and deeper analysis is missing.

**Quality:**

3

**Strengths And Weaknesses:**

Strengths:
1. This paper is well motivated. Since Vast Vocabulary Object Detection task have numeous classes, mining the sparse reward from positive categories is meaningful.
2. This paper is well written and easy to follow.
3. Experiments show its effectiveness, not only suporier in V3Det benchmarks, but also competitive in COCO dataset, comparable with strong baselines like Rank-DETR.

Weakness:
1. The Positive-to-negative gradient plot should be drawn in longer iterations.

---

> ### Author Rebuttal · Authors · 2025-07-31
>
> Thank you for your positive feedback and recognition of our work. We are deeply grateful for your positive assessment of our motivation and experimental results. Below, we address your concerns point-by-point.
>
> > Q1: The positive-to-negative gradient plot should be drawn in longer iterations.
> >
>
> A1: Thank you for this suggestion. We have extended our positive-to-negative gradient ratio analysis to 200k iterations to provide a more comprehensive view of the training dynamics.  Due to policy constraints that restrict the inclusion of images, we present the results in Table R-6 below, sampling key iterations throughout the training process.
>
> The extended analysis reveals several key insights: CQ-DINO maintains consistently higher positive-to-negative gradient ratios during the critical early training phase (less than 10k iterations), demonstrating effective gradient rebalancing when learning is most crucial. In contrast, DINO with Cross-Entropy loss exhibits severe positive gradient dilution in early iterations (starting at 0.01) before gradually recovering. While all methods eventually converge to similar gradient ratios (~1.1) in later training stages (>25k iterations), CQ-DINO achieves this balanced state much earlier, resulting in more efficient training.
>
> This extended analysis strongly supports our hypothesis that CQ-DINO effectively addresses the positive gradient dilution problem identified in vast vocabulary detection, particularly during the crucial early training phase where proper gradient balance is essential for learning meaningful representations.
>
> Table R-6:  Positive-to-negative gradient ratio across extended training iterations.
>
> | Iter | 0.5k | 1k | 1.5k | 2k | 2.5k | 3k | 5k | 7k | 10k | 15k | 25k | 50k | 75k | 100k | 200k |
> | --- | --- | --- | --- | --- | --- | --- | --- | --- | --- | --- | --- | --- | --- | --- | --- |
> | DINO w/CE | 0.01 | 0.09 | 0.11 | 0.06 | 0.11 | 0.28 | 0.43 | 0.62 | 0.79 | 0.93 | 1.00 | 1.10 | 1.04 | 1.12 | 1.10 |
> | DINO w/FL | 0.63 | 0.31 | 0.47 | 0.68 | 0.87 | 0.77 | 0.91 | 0.84 | 0.88 | 0.95 | 1.10 | 1.07 | 1.09 | 1.17 | 1.12 |
> | CQ-DINO | 1.08 | 1.56 | 1.21 | 1.64 | 1.70 | 1.39 | 1.50 | 1.23 | 1.12 | 1.18 | 1.12 | 1.19 | 1.16 | 1.08 | 1.14 |
>
> > Q2: Could the proposed CQ-DINO handle category correlations in datasets with incomplete or ambiguous hierarchies?
> >
>
> A2: Great question!  To evaluate CQ-DINO's robustness to incomplete hierarchical information, we conducted controlled experiments by artificially degrading the V3Det hierarchical structure in two ways:
>
> **1. Missing hierarchies**: We randomly removed hierarchical connections for 1/6 of the categories, simulating datasets with incomplete structural information.
>
> **2. Ambiguous hierarchies**: We introduced fuzzy/ambiguous hierarchical relationships for another 1/6 of the categories, representing scenarios where category boundaries are unclear.
>
> When hierarchical information is incomplete, CQ-DINO achieves 51.7% AP and outperforms the baseline without category correlations (51.1% AP) by 0.6% AP.  These results demonstrate that CQ-DINO can handle real-world scenarios where hierarchical structures are imperfect or incomplete. Besides, when hierarchical information is completely missing, CQ-DINO can seamlessly switch to implicit relation learning through the self-attention module (51.3% AP), which also outperforms the no-correlation baseline.
>
> Table R-7: Performance under incomplete hierarchical structures.
>
> | Method | $AP$ | $AP_{50}$ | $AP_{75}$ | $AR^c$ |
> | --- | --- | --- | --- | --- |
> | No correlation | 51.1 | 56.6 | 53.5 | 80.9 |
> | Self-Attention module | 51.3 | 56.7 | 53.7 | 75.5 |
> | Incomplete hierarchical tree | 51.7 | 57.1 | 54.0 | 82.47 |
> | Complete hierarchical tree | 52.3 | 57.7 | 54.6 | 83.3 |
>
> > Q3: Visualization and deeper analysis is missing.
> >
>
> A3: To provide deeper analysis beyond the positive-to-negative gradient ratio, we conducted additional experiments examining the hard negative gradient dilution problem discussed in Section 3.1. We define hard negatives as the top 10% of categories with the highest prediction scores among all negative categories, representing the most confusing negatives that require focused learning.
>
> Results depicted in Table R-8 show that our CQ-DINO consistently achieves the highest hard negative ratios throughout training, particularly in the critical early stages (less than 10k iterations), confirming our theoretical analysis (in Section 3.1) that image-guided query selection performs effective implicit hard negative mining.
>
>  Due to policy constraints that restrict the inclusion of images in this response, we will include comprehensive visualizations in the revised paper showing: (1) positive-to-negative gradient ratio plot. (2) Hard negative-to-all negative gradient ratio plot. (3) Visualization of detection performance on challenging samples and failure case analysis.
>
> Table R-8: Hard negative-to-all negative gradient ratio across training iterations
>
> | Iter | 0.5k | 1k | 1.5k | 2k | 2.5k | 3k | 5k | 7k | 10k | 15k | 25k | 50k | 75k | 100k | 200k |
> | --- | --- | --- | --- | --- | --- | --- | --- | --- | --- | --- | --- | --- | --- | --- | --- |
> | DINO w/CE | 0.23 | 0.53 | 0.61 | 0.67 | 0.72 | 0.77 | 0.89 | 0.97 | 0.99 | 0.99 | 1.00 | 1.00 | 1.00 | 1.00 | 1.00 |
> | DINO w/FL | 0.69 | 0.65 | 0.63 | 0.62 | 0.64 | 0.66 | 0.81 | 0.88 | 0.93 | 0.97 | 0.99 | 0.99 | 0.99 | 0.99 | 0.99 |
> | CQ-DINO | 0.83 | 0.90 | 0.89 | 0.91 | 0.92 | 0.93 | 0.94 | 0.96 | 0.98 | 0.98 | 0.98 | 0.99 | 0.99 | 0.99 | 0.99 |

---

> > ### Comment · Reviewer_8AUD · 2025-08-04
> >
> > I thank the authors for their thorough rebuttal and additional analysis, which has resolved the majority of my concerns. Accordingly, I decide to keep my rating as borderline accept.

---

> > > ### Author Response · Authors · 2025-08-04
> > >
> > > Thank you again for your valuable suggestions and insightful comments. We are glad that we have addressed your concerns. We are greatly encouraged by your acceptance. Please feel free to reach out if you have any further questions.

---

### Official Review · Reviewer_YWme · 2025-07-03

**Clarity:** 3
**Significance:** 2
**Originality:** 3
**Rating:** 4
**Confidence:** 4

**Summary:**

This paper investigates scaling object detectors to vast vocabularies. The authors identify and analyze two key issues with standard classification-based methods: "positive gradient dilution," where positive samples are overwhelmed by the sheer number of negatives, and "hard negative gradient dilution," where informative negative gradients are lost among a sea of easy ones. To address this, they propose CQ-DINO, a novel framework that reformulates classification as a contrastive task. The core of their method is an image-guided query selection module that dynamically selects a small subset of relevant "category queries" for each image. This pruning of the category space aims to rebalance gradients and perform implicit hard negative mining. The selected queries are then used in a DINO-like architecture to produce detections. The authors demonstrate state-of-the-art results on the vast-vocabulary V3Det benchmark.

**Questions:**

1.  The category recall of 83.3% appears to be a bottleneck. Have you analyzed the characteristics of the ~17% of objects whose categories are missed by the selection module? Are they primarily rare categories, visually similar categories, or small objects? Understanding this failure mode is critical, as my evaluation of the method's robustness hinges on whether this ceiling can be raised significantly.
2.  While CQ-DINO outperforms a tuned Focal Loss baseline by 4.0% AP, it is also a much more complex system. How can we be sure that the performance gain is due to a fundamentally better approach to gradient balancing, rather than a side effect of adding more complexity, parameters, and a separate supervision signal (the selection loss) to the overall training pipeline?

**Ethical Concerns:**

["NO or VERY MINOR ethics concerns only"]

**Final Justification:**

The author's rebuttal has solved most of my concerns. Hence, I decided to raise my score to borderline accept.

**Limitations:**

Yes

**Paper Formatting Concerns:**

I did not notice any major formatting issues.

**Quality:**

3

**Strengths And Weaknesses:**

Strengths:
1.  The paper provides an insightful analysis of the gradient dilution problem, offering a clear empirical motivation for why existing methods struggle with vast vocabularies.
2.  The main idea of using learnable category queries and an image-guided selection mechanism to dynamically prune the search space is clever. It directly targets the identified problem of an overwhelming number of negative classes.
3. The experimental results on the challenging V3Det benchmark are strong, showing a clear improvement over previous methods, including a well-tuned DINO baseline. This demonstrates the empirical effectiveness of the proposed approach.

Weakness:
1. The framework is quite complex and relies on a two-stage training process: a pre-training stage for the selection module followed by a full-pipeline fine-tuning stage. As the authors acknowledge in their limitations, this can lead to suboptimal coordination between the stages and makes the model more difficult to train and deploy compared to a true end-to-end solution.
2.  While the analysis of "gradient dilution" is thorough, the underlying issue is a severe case of the well-known class imbalance problem. Focal Loss was designed to address the dominance of easy negatives. The paper shows their method works better than tuning Focal Loss, but the problem framing itself is not entirely new.
3. The paper heavily utilizes components from Grounding DINO for the feature enhancement and decoding stages. As such, the contribution can be seen less as a completely new detection paradigm and more as a sophisticated "pre-filtering" front-end for an existing powerful detector.

---

> ### Author Rebuttal · Authors · 2025-07-31
>
> Thank you for thoroughly reading our paper. We are deeply grateful that you recognize the significance of our gradient dilution analysis. We are very thankful that you find our method to be a clever solution to the identified gradient dilution problems. We address your concerns below.
>
> > Q1: Complexity of our two-stage training approach.
> >
>
> A1: Indeed, the two-stage training introduces some additional complexity. In practice, the overhead is minimal. Stage 1 requires only ~1 hour compared to the 57 hours for full training. For deployment, only the final trained model is needed without two-stage training. For future work, we are actively exploring methods such as progressive query selection to develop stable end-to-end alternatives to keep high category recall throughout training.
>
> > Q2: Gradient dilution issue VS class imbalance problem.
> >
>
> A2: We appreciate the reviewer's insight connecting gradient dilution to class imbalance. However, our work identifies and addresses a distinct scaling challenge in vast vocabularies that differs from traditional class imbalance problems.
>
> Traditional class imbalance mainly focuses on sample distribution disparities between rare and common classes. In contrast, gradient dilution in vast vocabularies affects even common classes due to the overwhelming negative space created by thousands of categories. Our theoretical analysis (Equations 2-4) demonstrates that the positive-to-negative gradient ratio scales as $\rho \propto \frac{1}{C\cdot ε}$, where $C$ is the vocabulary size and ε represents the average activation probability of negative classes. This means the problem severity increases proportionally with vocabulary size $C$, not just sample imbalance. Moreover, vast vocabularies cause informative hard negatives to be overwhelmed by numerous easy negatives.
>
> To address the concern about whether CQ-DINO can alleviate the class imbalance problem in V3Det, we evaluate the results of rare classes (number of training samples less than 10) and common classes (otherwise) in Table R-3. The results show that even with carefully tuned hyperparameters, Focal Loss cannot fundamentally solve the 1/C scaling problem. Our dynamic category selection directly addresses the optimization challenges, achieving consistent improvements across both rare and common categories.
>
> Table R-3: Performance of rare classes and common classes on V3Det dataset.
>
> | Method | Rare | Common | All |
> | --- | --- | --- | --- |
> | DINO-SwinB-22K | 36.4 | 43.8 | 43.4 |
> | DINO-SwinB-22K (Focal α=0.50, γ=2) | 42.1 | 48.4 | 47.4 |
> | CQ-DINO-SwinB-22K | 45.3 | 53.3 | 52.3 |
> | DINO-SwinL | 45.2 | 51.0 | 48.5 |
> | DINO-SwinL (Focal α=0.50, γ=2) | 45.5 | 51.7 | 50.1 |
> | CQ-DINO-SwinL | 46.9 | 54.1 | 53.0 |
>
> > Q3: Is this just pre-filtering for existing Grounding DINO detector?
> >
>
> A3: No, CQ-DINO is fundamentally different from pre-filtering. It addresses the optimization challenges that Grounding DINO cannot solve through architectural modifications alone.
>
> Grounding DINO faces scalability bottlenecks for vast vocabulary detection (Section 2.2, lines 115-119). With its 128 token limit per inference pass, detecting all 13,204 V3Det categories requires **over 331 sequential inference passes per image**. For the complete V3Det validation set (30k images), this requires approximately **768 GPU hours** on H800 GPUs for inference alone, which is almost infeasible with our limited computational resources.
>
> While we leverage Grounding DINO's proven feature enhancer and decoder (which excel at fusing image features and category queries), our baseline performance is nearly identical to the optimally Focal loss tuned DINO (47.3% Table3 row 1 vs 47.4% Table 9 row5). This further confirms that architectural components alone cannot solve vast vocabulary challenges.
>
> > Q4:  What characterizes the 17% of missed objects causing the category recall bottleneck - rare categories, small objects, or visual similarity?
> >
>
> A4: We analyze the 3,527 categories with recall lower than 83.3% to understand the characteristics of missed objects. Our analysis reveals that both rare category performance and small object detection represent significant bottlenecks compared to our overall performance of 52.3% AP.
>
> (1) **Category frequency analysis (Table R-4):** Among the low-recall categories, 374 are rare categories and 3,153 are common categories. On these challenging categories, CQ-DINO achieves 20.5% AP on rare categories, substantially lower than our overall 52.3% AP, indicating that rare categories face a significant performance gap.
>
> (2)  **Object scale analysis (Table R-5):** The most significant performance gaps occur for small objects, where CQ-DINO achieves only 14.1% AP compared to our overall performance of 52.3% AP. This indicates that object scale is the primary factor in the recall bottleneck, with small objects being particularly challenging.
>
> While we do not achieve strong absolute performance in these challenging cases, CQ-DINO still consistently outperforms both baseline DINO and DINO using carefully tuned focal loss across all categories and scales. This demonstrates that **these are common difficulties in vast vocabulary detection** rather than unique limitations of our approach.
>
> Table R-4:  Analysis on low recall categories.
>
> | Method | Rare | Common | All |
> | --- | --- | --- | --- |
> | DINO-SwinB-22K | 15.5 | 27.1 | 25.8 |
> | DINO-SwinB-22K (Focal α=0.50, γ=2) | 19.9 | 31.1 | 29.9 |
> | CQ-DINO-SwinB-22K | 20.5 | 32.1 | 30.9 |
>
> Table R-5: Analysis on object scales.
>
> | Method | Small | Middle | Large |
> | --- | --- | --- | --- |
> | DINO-SwinB-22K | 9.1 | 16.5 | 33.8 |
> | DINO-SwinB-22K (Focal α=0.50, γ=2) | 11.6 | 20.2 | 38.3 |
> | CQ-DINO-SwinB-22K | 14.1 | 23.1 | 38.9 |
>
> > Q5: Does CQ-DINO's 4.9% AP improvement stem from gradient balancing or increased model complexity?
> >
>
> A4: The performance gain is fundamentally owing to the gradient balancing, not model complexity. We provide comprehensive evidence through detailed ablation studies:
>
> (1) Our baseline without image-guided query selection (47.3% AP in Table 4, row 1) achieves nearly identical performance to DINO trained with optimally-tuned Focal Loss (47.4% AP in Table 9). This demonstrates that the model architecture itself does not provide inherent advantages.
>
> (2) Table 4 shows that adding image-guided query selection alone improves AP from 47.3% to 51.1% (+3.8%), which directly addresses gradient dilution by reducing the negative search space. Table R-1 shows that training-stage gradient dilution mitigation alone (row 4) provides +1.9% AP. This further validates that gradient rebalancing is the core mechanism.
>
> (3) Encoding category correlations contributes +1.2% AP in Table 4 (52.3% vs 51.1%, row 6 vs row 5).
>
> Table R-1. Additional ablations on isolated component contributions. 0 indicates the component is disabled and 1 indicates it is enabled.
>
> | Training-stage gradient dilution | Inference-stage category selection | Encoding category correlations | V3Det AP (SwinB-22K) | V3Det FPS | COCO AP (SwinL) | COCO FPS |
> | --- | --- | --- | --- | --- | --- | --- |
> | 0 | 0 | 0 | 47.3 | 0.7 | 57.5 | 9.8 |
> | 0 | 0 | 1 | 49.4 | 0.6 | 57.9 | 9.7 |
> | 0 | 1 | 1 | 46.6 | 10.6 | 58.2 | 9.9 |
> | 1 | 0 | 1 | 51.3 | 0.6 | 58.2 | 9.7 |
> | 1 | 1 | 0 | 51.1 | 10.8 | 58.3 | 10.0 |
> | 1 | 1 | 1 | 52.3 | 10.6 | 58.5 | 9.9 |

---

> > ### Comment · Reviewer_YWme · 2025-08-05
> >
> > Thanks for the authors' rebuttal.
> > For A2, I am still not convinced. The core of the class imbalance and the proposed "gradient dilution" is the same, both reflecting the issue that model receives too much supervision from one side (either vast negatives or common classes) and hurts the performance of other side (rare positive or rare classes). The technique to solve the issue should be also similar, eg just increasing the portion of "rare" side or decreasing the portion of "common" side. I do not see fundamental differences.
> > For A3,  I still do not understand why it is not adding a pre-filtering to grounding dino, it seems to me exactly that. (as shown in figure 1)
> >
> > I thank again the authors for their thorough rebuttaI but i currently decide to keep my rating.

---

> > > ### Author Response · Authors · 2025-08-06
> > > **Further clarifications on Q2 and Q3 (part 1)**
> > >
> > > > Response to comment Q2:
> > > >
> > >
> > > We thank the reviewer for this insightful follow-up question. We totally agree with you that the two problems are deeply related, and their core symptom—the suppression of learning signals from minority groups (rare classes or positive samples)—is the same. However, we respectfully argue that there is an **important difference in the source of the problem**, which provides **new insights into the vast vocabulary task** and necessitates a more sophisticated solution.
> > >
> > > 1.**Identifying severe gradient dilution due to the vast size $C$ as the major cause of unsatisfactory performance in vast vocabulary object detection**
> > >
> > > In the original paper's analysis (Sec 3.1), we intentionally assume a relatively balanced data distribution. By assuming $n_{c^+}$ is constant, our original formula $\rho \propto \frac{1}{C\cdot ε}$ demonstrates that the positive gradient signal is diluted by a factor proportional to $C$.
> > >
> > > A more general gradient ratio $\rho_{c^+}$ for the positive class $c^+$ can be expressed as:
> > >
> > > $$
> > > \rho_{c^+} = \frac{n_{c^+} \cdot ||\nabla_{z_{c^+}}\mathcal{L}||}{\sum_{c^-\neq c^+}^C n_{c^-}\cdot||\nabla_{z_{c^-}}\mathcal{L}||} \approx \frac{n_{c^+}\cdot(1-\sigma(z_{c^+}))}{ \mathbb{E}[\sigma(z_{c^-})]\cdot\sum_{c^-\neq c^+}^C n_{c^-}} \propto \frac{n_{c^+}\cdot \epsilon^+}{(N-n_{c^+})\cdot \epsilon^-} ,
> > > $$
> > >
> > > where $n_{c^+}$ is the sample count for the positive class. $N$ is the total sample count. $\epsilon^+$ and $\epsilon^-$ represent the average gradient magnitudes for positive and negative samples, respectively.
> > >
> > > This formula reveals two distinct sources of gradient dilution:
> > >
> > > - **The Class Imbalance:** This is primarily reflected by the term $n_{c^+}$ in the numerator. When a class is rare, $n_{c^+}$ is small, directly reducing the ratio $\rho_{c^+}$.
> > > - **The Vast Vocabulary Size:** This is primarily driven by the term $N - n_{c^+}$ in the denominator. In vast vocabulary settings ($C>10,000$), $N - n_{c^+}$ becomes enormous because it aggregates all negative samples from the other $C-1$ categories. The large $C$ further reduces the ratio $\rho_{c^+}$, making the gradient dilution more severe in vast vocabulary object detection. This impacts **both rare and common classes**.
> > >
> > > 2.**Comparing CQ-DINO with traditional class-imbalanced methods in addressing gradient dilution**
> > >
> > > Traditional class-imbalanced methods address gradient dilution as follows:
> > > - **Re-sampling:** These techniques aim to artificially boost the contribution of the numerator, $n_{c^+}$.
> > > - **Re-weighting  (e.g., Focal Loss):** These methods focus on reducing $\epsilon^-$ by down-weighting easy negatives or increasing $\epsilon^+$ by up-weighting positives.
> > >
> > > While effective in low-to-medium vocabulary settings, these methods struggle when $C$ is vast.  Our experiments (Table 9, Table R-3) confirm this, showing that even a carefully tuned Focal Loss cannot fully bridge the performance gap.
> > >
> > > CQ-DINO addresses the problem from a different angle. Instead of just tweaking the terms in the ratio, our image-guided query selection **strategically reduces the size of the classification problem itself**. It prunes the $C$ categories down to a small, relevant subset $C’$ ($C’ \ll C $). This changes the optimization landscape from: "Balancing 1 positive against $C-1$ negatives" to "Balancing 1 positive against $C’-1$ hard negatives".
> > >
> > >
> > > We really appreciate your responsible and professional comments. We also would like to thank you for the opportunity to discuss our work further, which greatly helps us to clarify the contributions of our work: 1) We are the first, to our knowledge, to discover severe gradient dilution due to the vast size $C$ as the major cause of unsatisfactory performance for vast vocabulary object detection, which provides new insights into this task; 2) We develop a sophisticated image-guided query selection module to mitigate gradient dilution during training, which is more effective in addressing gradient dilution than traditional class-imbalanced methods. We will clarify them in the revised paper, and hope these clarifications can potentially address your concerns.

---

> > > ### Author Response · Authors · 2025-08-06
> > > **Further clarifications on Q2 and Q3 (part 2)**
> > >
> > > > Response to comment Q3:
> > > >
> > > We sincerely thank the reviewer for this question, which allows us to clarify the core methodological contribution of CQ-DINO.
> > >
> > > We understand why our method might appear as a "pre-filtering" step, especially from a high-level, inference-flow perspective (as depicted in Figure 1). Indeed, at inference time, the image-guided query selection module does perform as pre-filtering that filters out irrelevant categories to accelerate inference and reduce the negative space. We believe, however, that its most significant contribution lies not in this inference-time behavior, but in mitigating the important training-time gradient dilution problem.
> > >
> > > To explore this distinction, we considered what would happen if the module were only an inference-stage filter. In such a case, we train on all categories and only apply the selection at inference. As depicted in Table R-1, when we isolate the selection to the inference stage, performance drops significantly from 52.3% AP to 46.6% AP. This suggests that the module's primary benefit lies in the training process, where it actively rebalances gradients by focusing on a sparse set of relevant positive and hard-negative categories.
> > >
> > > From a training perspective, a straightforward solution is indeed applying a pre-filtering step to Grounding DINO. Therefore, we also conducted alternative experiments with pre-filtering that explicitly decouple selection and detection, as detailed in Table R-9.
> > >
> > > - The “Frozen Selection” baseline pre-trains all encoder components for query selection as pre-filtering, then freezes the shared encoder during detection training.
> > > - The “Dual-Encoder” method improves on the "Frozen Selection" baseline by using an additional trainable encoder for detection training.
> > >
> > > Our end-to-end CQ-DINO (52.3% AP) substantially outperforms the "Frozen Selection" baseline by 1.8% AP. This performance gap primarily stems from the frozen encoder's inability to adapt to the detection task. The "Dual-Encoder" approach (51.9% AP) addresses this limitation by allowing the additional encoder to continue training for detection while using a well-trained selection model for pre-filtering. Though the “Dual-Encoder” achieves similar performance to our CQ-DINO, this comes at a significant computational cost—approximately 25% slower inference and 40% higher GPU memory consumption. Such a dual-encoder design essentially doubles the encoding overhead, making it impractical for real-world deployment. While using a lightweight query selection model could mitigate these costs, it may degrade pre-filtering capability and ultimately the performance.
> > >
> > > In light of this evidence, while CQ-DINO may benefit from pre-filtering mechanism at inference, its core advantage lies in the jointly-optimized, training-time mechanism for mitigating severe gradient dilution thanks to our discovery. Though the straightforward combination of simple pre-filtering and Grounding DINO may also achieve similar performance, our CQ-DINO still performs slightly better while being more efficient. We really thank you for helping us to better clarify this, which will be incorporated in the revised paper. If any aspect remains unclear, we look forward to any additional comments or questions.
> > >
> > > Table R-1. Additional ablations on isolated component contributions.
> > >
> > > | Training-stage gradient dilution | Inference-stage category selection | Encoding category correlations | V3Det AP (SwinB-22K) | V3Det FPS | COCO AP (SwinL) | COCO FPS |
> > > | --- | --- | --- | --- | --- | --- | --- |
> > > | 0 | 0 | 0 | 47.3 | 0.7 | 57.5 | 9.8 |
> > > | 0 | 0 | 1 | 49.4 | 0.6 | 57.9 | 9.7 |
> > > | 0 | 1 | 1 | 46.6 | 10.6 | 58.2 | 9.9 |
> > > | 1 | 0 | 1 | 51.3 | 0.6 | 58.2 | 9.7 |
> > > | 1 | 1 | 0 | 51.1 | 10.8 | 58.3 | 10.0 |
> > > | 1 | 1 | 1 | 52.3 | 10.6 | 58.5 | 9.9 |
> > >
> > > Table R-9: Comparison with alternative selection strategies.
> > >
> > > | Method | AP | $AR^C$ | FPS | Training memory |
> > > | --- | --- | --- | --- | --- |
> > > | Frozen Selection | 50.5 | 85.7 | 10.6 | 28.2GB |
> > > | Dual-Encoder | 51.9 | 85.7 | 7.9 | 46.7GB |
> > > | CQ-DINO (End-to-end) | 52.3 | 83.3 | 10.6 | 33.3GB |

---

> > > > ### Author Response · Authors · 2025-08-08
> > > > **Last day of discussion**
> > > >
> > > > Dear Reviewer YWme,
> > > >
> > > >  We sincerely appreciate your professional and insightful comments. We are very encouraged that you recognized the significance of our gradient dilution analysis, finding our method a "clever solution" and our results "strong."
> > > >
> > > > As the discussion period is drawing to a close, we would kindly appreciate if you could follow up on our response. Your input is highly valued, and will undoubtedly help us improve our work. We are open to further discussions to clarify any remaining questions or concerns. We hope our responses have been helpful and would be grateful for your consideration in re-evaluating our paper.
> > > >
> > > >  Thank you again for your time and consideration.
> > > >
> > > >  Sincerely,
> > > >
> > > >  The Authors

---

> > > > > ### Author Response · Authors · 2025-08-08
> > > > >
> > > > > Dear Reviewer YWme,
> > > > >
> > > > > As the discussion phase ends today, we will not be able to further clarify potential additional concerns. We would be very grateful if you could respond to our further comment and offer us an opportunity to address any questions you might have! Your feedback is incredibly valuable to us, and we sincerely hope that our responses have addressed your concerns.
> > > > >
> > > > > Thank you again for your time and consideration!
> > > > >
> > > > > Sincerely,
> > > > >
> > > > > The Authors

---

### Official Review · Reviewer_BueR · 2025-07-05

**Clarity:** 3
**Significance:** 3
**Originality:** 2
**Rating:** 4
**Confidence:** 4

**Summary:**

This paper studies gradient dilution issues in vast vocabulary object detection arising from numerous categories: (1) gradients from negative categories overwhelm those from positive categories; (2) within negative categories, gradients from easy samples dominate. To mitigate this, unlike standard classifiers employing static, image-irrelevant category embeddings, the proposed method utilizes image-adaptive category queries to perform image-level category selection prior to instance-level classification.

Specifically, these category queries aggregate image features via cross-attention and perform image-level classification to filter out absent categories. This significantly reduces the number of categories (e.g., from over 13,000 to 100) for subsequent instance-level classification, thereby alleviating the described gradient imbalance. To handle category hierarchies, the method leverages inherent tree structures to construct hierarchical category queries with an adaptive weighting mechanism.

When integrated into Grounding DINO, the proposed approach achieves 53.0 AP on the V3Det dataset, outperforming prior methods. A slight performance gain is also observed on the COCO dataset.

**Questions:**

I assign a borderline rating due to two primary concerns:
* Lack of discussion on prior works exploring category query techniques (detailed in Weakness 1),
* Insufficient validation of the core claim regarding training-stage gradient dilution mitigation (detailed in Weakness 3).

I am open to increasing my rating if the authors address these issues as suggested.

**Ethical Concerns:**

["NO or VERY MINOR ethics concerns only"]

**Final Justification:**

I thank the authors for their rebuttal, which includes the discussion and experiments as suggested. The rebuttal has addressed my primary concerns:
* The lack of related work discussion has been resolved by adding a dedicated section on Category Query-based methods, which clarifies the paper's distinct motivation from these approaches.
* The insufficient ablation studies have been supplemented with Table R-1, isolating the impacts of inference-stage category selection and training-stage gradient dilution mitigation. This demonstrates that performance gains mainly stem from the latter, thereby supporting the central claim.

Overall, I acknowledge the paper's contributions: the identification and analysis of the gradient dilution problem, an effective solution, and SOTA performance. Therefore, I am voting for acceptance.

**Limitations:**

yes

**Quality:**

2

**Strengths And Weaknesses:**

Strengths:
1. The writing is clear.
2. The proposed method achieves 53.0 AP on the V3Det dataset, outperforming previous methods including DINO and PROVA by a significant margin. This demonstrates effectiveness for vast vocabulary object detection.

Weaknesses:
1. Lack of discussion on prior works: The core ideas of this paper—(1) modeling classification categories in an image-adaptive manner instead of using static, image-irrelevant embeddings, and (2) leveraging image-level classification prior to dense predictions—have been explored in existing works such as Query2Label [1], RankSeg [2], and CQL [3]. Query2Label pioneered the concept of category queries. Both RankSeg and CQL employ category queries for image-level classification to enhance dense prediction tasks (semantic segmentation and human-object interaction detection, respectively). RankSeg and CQL both utilize category queries for image-level classification to aid dense prediction tasks (semantic segmentation and human-object interaction detection). RankSeg retains only the top-k semantic categories, a strategy highly similar to this paper and shown to improve accuracy. CQL does not explicitly discard low-confidence categories but integrates image-level classification scores into instance-level scores, downweighting instances from low-confidence categories. However, these works are not discussed. I recommend adding a dedicated section on category query techniques in the related work.
2. Incomplete baseline comparison: The proposed CQ-DINO is built upon the Grounding DINO framework, yet only DINO is used as the baseline. To isolate the contributions of the proposed method versus the Grounding DINO framework, comparative experiments are necessary.
3. Insufficient validation of core claims:
    * Prior works (e.g., RankSeg, CQL) demonstrate that image-level category selection improves inference-stage accuracy for dense prediction tasks by filtering or downweighting low-confidence categories. This paper, however, focus solely about its training-stage impact (mitigating gradient dilution) without addressing inference benefits.
    * The gradient dilution issue may be largely alleviated by focal loss. As shown in Figure 2, the positive-to-negative gradient ratio is near 0 with CE loss but rises to ~0.5 with focal loss. While CQ-DINO achieves a ratio of ~1.5, the difference is marginal (several-fold rather than orders of magnitude) and potentially addressable through hyper-parameter tuning. Section 4.4 demonstrates that tuning hyper-parameters of focal loss effectively improves vanilla DINO’s performance. Though still below CQ-DINO, it remains unclear whether the performance gap stems from: (a) the Grounding DINO framework, (b) inference-stage category selection, or (c) mitigation of training-stage gradient dilution.
    * I recommend adding ablations isolating contributions from: (a) Grounding DINO, (b) inference-stage category selection, and (c) training-stage gradient dilution mitigation. This is essential to validate the paper’s central claim.

[1] Query2Label: A Simple Transformer Way to Multi-Label Classification

[2] RankSeg: Adaptive Pixel Classification with Image Category Ranking for Segmentation

[3] Category Query Learning for Human-Object Interaction Classification

---

> ### Author Rebuttal · Authors · 2025-07-31
>
> Thank you for your insightful comments and kind words. We greatly appreciate your recognition of our significant performance improvements. Below we will address your concerns and provide further clarification.
> > Q1: Lack of discussion on prior works.
> >
> A1: Thank you for your valuable feedback regarding the relevant literature [2, 8, 10], particularly the similarity between RankSeg's top-k category selection and our image-guided query selection. We will add a dedicated "Category Query-based Methods" section to our related work.
>
> In the following, we respectfully clarify several key distinctions that differentiate our approach:
>
> 1. Novel theoretical motivation:  To the best of our knowledge, our work provides the first systematic analysis of gradient dilution challenges in vast vocabulary detection. This theoretical foundation directly motivates our image-guided query selection design. In contrast, RankSeg employs top-k selection primarily for reducing the negative sample space during inference, which decreases computational cost and improves accuracy.
> 2. Scale-specific challenges: While top-k selection during inference works for semantic segmentation with limited vocabularies, vast vocabularies present challenges. Our analysis in Table R-1 reveals: inference-stage category selection harms performance in V3Det (vast vocabulary: 49.4% → 46.6%, rows 2 → 3) while helping in COCO (limited vocabulary: 57.9% → 58.2%). This demonstrates the necessity of addressing gradient dilution during training, not just inference optimization.
> 3. Hierarchical relationship modeling: Unlike previous works operating on relatively flat category structures, we specifically address the complex hierarchical relationships inherent in vast vocabulary datasets through our novel hierarchical tree construction method.
>
> Table R-1. Additional ablations on isolated component contributions.
> | Training-stage gradient dilution | Inference-stage category selection | Encoding category correlations | V3Det AP (SwinB-22K) | V3Det FPS | COCO AP (SwinL) | COCO FPS |
> | --- | --- | --- | --- | --- | --- | --- |
> | 0 | 0 | 0 | 47.3 | 0.7 | 57.5 | 9.8 |
> | 0 | 0 | 1 | 49.4 | 0.6 | 57.9 | 9.7 |
> | 0 | 1 | 1 | 46.6 | 10.6 | 58.2 | 9.9 |
> | 1 | 0 | 1 | 51.3 | 0.6 | 58.2 | 9.7 |
> | 1 | 1 | 0 | 51.1 | 10.8 | 58.3 | 10.0 |
> | 1 | 1 | 1 | 52.3 | 10.6 | 58.5 | 9.9 |
>
> We will incorporate the following related work section:
>
> **Category Query-based Methods**
>
> The paradigm of learnable queries emerged in computer vision with DETR [1], demonstrating a shift from fixed components to learnable representations that capture task-specific patterns. DETR's query mechanism inspired similar strategies across diverse tasks, including classification [2], segmentation [3], and multimodal learning [4].
>
> Among these developments, category queries represent an innovation introduced by Query2Label [2]. Rather than relying on fixed classification heads, Query2Label proposed learnable category embeddings to capture category-specific features. Subsequent works such as ML-Decoder [5] have outperformed conventional classification methods. The effectiveness of category queries in classification motivated their extension to dense prediction tasks. CQL [7] employs category queries for human-object interaction classification. Similarly, ControlCap [8] leverages category queries to provide semantic guidance for region captioning tasks. RankSeg [9] integrates category queries into semantic segmentation by performing pixel-level classification while dynamically selecting only the top-k most relevant categories during inference. This selective approach reduces the negative search space, improving both computational efficiency and segmentation accuracy.
>
> While prior works have explored category queries in various contexts, our work addresses a fundamentally different challenge specific to vast vocabulary scenarios. We provide the first, to-our-knowledge, systematic theoretical analysis of gradient dilution issues that arise when dealing with vast category vocabularies, which motivates our image-guided query selection design. Additionally, unlike previous methods working with limited category scales, we specifically tackle the complex hierarchical relationships inherent in vast vocabulary datasets through our hierarchical tree construction method, designed to encode category correlations across thousands of semantically related categories.
>
> Reference
>
> [1] End-to-End Object Detection with Transformers
>
> [2] Query2Label: A Simple Transformer Way to Multi-Label Classification
>
> [3] Per-Pixel Classification is Not All You Need….
>
> [4] BLIP-2: Bootstrapping Language-Image….
>
> [5] ML-Decoder: Scalable and Versatile Classification Head
>
> [6] Category Query Learning for Human-Object Interaction Classification
>
> [7] ControlCap: Controllable Region-level Captioning
>
> [8] RankSeg: Adaptive Pixel Classification with Image Category Ranking for Segmentation
> > Q2: Incomplete baseline comparison with Grounding DINO.
> >
> A3: As suggested, we provide the following clarification. Detailed in lines 115-119, Grounding DINO faces scalability limitations for vast vocabulary detection. With a token limit of 128 per inference pass and V3Det's 13204 categories, detecting all categories requires over 331 sequential inference passes per image. For the full V3Det val set (30k images), this translates to approximately **768 GPU hours on H800 GPUs** for inference alone, making comprehensive evaluation computationally prohibitive.
>
> To provide meaningful comparison while addressing computational constraints, we evaluate on V3Det's rare classes in Table R-2. This subset represents the most challenging categories where gradient dilution effects are most pronounced. Grounding DINO is a pretrained model from mmdetection, trained on O365, GoldG, and V3Det datasets. DINO and CQ-DINO are trained on the V3Det dataset.
>
> Table R-2: Performance on V3Det rare classes (1072 categories).
> | Method | AP |
> | --- | --- |
> | DINO | 36.4 |
> | Grounding DINO | 23.4 |
> | CQ-DINO | 46.9 |
>
> Grounding DINO underperforms DINO (-13.0%), revealing limitations in its training paradigm in vast vocabulary settings. Grounding DINO uses positive labels with only randomly sampled small sets of negative labels during training. With few similar categories sampled during training, Grounding DINO fails to learn fine-grained distinctions between semantically similar categories—the core challenge in vast vocabulary detection.
>
> To isolate Grounding DINO’s component impacts, we conduct ablation studies in Table 4 (row 1) and Table R-1 (row 1). Substantial improvements show our method focuses on addressing the fundamental challenges of vast vocabulary detection rather than simply benefiting from architectural choices.
> > Q3: Adding ablations isolating contributions from: (a) Grounding DINO, (b) inference-stage category selection, and (c) training-stage gradient dilution mitigation.
> >
> A2: Thank you for this insightful suggestion. We now provide additional ablation studies in Table R-1 to isolate the contributions of each component and validate our main claim about gradient dilution mitigation.
>
> Component Definitions:
> (a) Grounding DINO components: The baseline architecture incorporating feature enhancer and decoder components from GroundingDINO (row 1 in Table R-1).
>
> (b) Inference-stage category selection:  Selecting only top-K relevant categories based on similarity of category queries and image features during inference.
>
> (c) Training-stage gradient dilution mitigation: Using image-guided query selection during training to mitigate gradient dilution.
>
> Key observations from Table R-1:
>
> (1) Training-stage gradient dilution mitigation is the primary contributor. On V3Det, this component alone provides a 1.9% AP improvement (comparing rows 2 and 4). This validates our hypothesis that gradient dilution is a challenge in vast vocabulary detection. On COCO, the improvement is modest (0.3% AP) since gradient dilution is less severe with only 80 categories.
>
> (2) Inference-stage category selection has different effects with respect to vocabulary size.
>
> • V3Det: Inference selection without gradient dilution mitigation actually hurts performance (rows 2 vs 3), demonstrating that simply reducing categories at inference cannot solve the underlying gradient dilution problem.
>
> • COCO: Inference selection provides modest improvements as expected when the negative space is reduced.
>
> (3) The combination achieves optimal performance and efficiency. When both training-stage gradient dilution mitigation and inference-stage selection are combined (row 6), we achieve the best results on both datasets. Inference-stage selection improves inference speed (0.6 → 10.6 FPS on V3Det).
>
> These additional ablation studies further demonstrate that (c) training-stage gradient dilution mitigation is the core contribution, while (a) Grounding DINO components provide the architectural foundation and (b) inference-stage category selection offers computational efficiency. Table R-1 will be incorporated into Table 4 in the revised paper.
> > Q4: Are Focal loss and CQ-DINO equally effective in solving gradient dilution problems in vast vocabulary detection?
> >
> No, CQ-DINO and Focal Loss address gradient dilution problems from different aspects. Our theoretical analysis in Sec. 3.1 reveals that the positive-to-negative gradient ratio scales as ρ ∝ 1/(C·ε), where C is the vocabulary size and ε represents the average activation probability of negative classes.
>
> While Focal Loss can mitigate gradient dilution by adjusting its parameters to reduce ε, it cannot address the core 1/C scaling problem inherent to vast vocabulary detection. As vocabulary size C grows beyond 10⁴ categories, this scaling factor becomes the dominant bottleneck regardless of ε optimization. In contrast, CQ-DINO fundamentally solves the scaling issue by reducing the effective vocabulary size from C to C' (where C' ≪ C) through image-guided query selection.

---

> > ### Author Response · Authors · 2025-08-08
> > **Last day of discussion**
> >
> > - Dear Reviewer BueR,
> >
> >     We sincerely appreciate your professional and insightful comments. We are very grateful for your recognition of our significant performance improvements. Due to the rush in finalizing the writing, some aspects may be missing or confusing. We have addressed these issues in our response and provided further elaboration.
> >
> >     With the discussion period nearing its end, we kindly request your acknowledgment of our response. Your expert feedback has been invaluable in strengthening our work. We value your insights and would be grateful if you could join the discussion to clarify any remaining questions or concerns. We hope our responses have fully addressed your concerns and would appreciate your consideration in re-evaluating our work.
> >
> >     Thank you again for your time and consideration.
> >
> >     Sincerely,
> >
> >     The Authors

---

> > > ### Author Response · Authors · 2025-08-08
> > >
> > > Dear Reviewer BueR,
> > >
> > > As the discussion period concludes today, we wanted to respectfully follow up and check if you had any final questions regarding our responses. We would be very grateful if you could acknowledge our response and share any further thoughts or clarifications you might have. Your feedback is incredibly valuable to us, and we sincerely hope that our responses have addressed your concerns.
> > >
> > > Thank you again for your time and consideration!
> > >
> > > Best,
> > >
> > > Authors

---

### Author Response · Authors · 2025-08-09
**Rebuttal Summary for Paper 3386  (Thanks to all the reviewers for your attention to CQ-DINO)**

Dear Area Chair,

Thank you for your time and for handling the review process for our paper. We are grateful for the reviewers’ insightful feedback, which has helped us significantly strengthen our work. We are greatly encouraged that the reviewers acknowledged the novelty and effectiveness of our work, including its **strong performance (R BueR, R YWme, R 8AUD, R 9iMy)**, **well-motivated and intuitive idea (R YWme, R 8AUD, R 9iMy)**, **insightful analysis of gradient dilution (R YWme)**, and **clear writing (R BueR, R 8AUD)**.

During the discussion period, we provided substantial new experiments, analyses, and visualizations to address all concerns. We are grateful that our responses addressed the questions from reviewers **R 8AUD** and **R 9iMy**. We understand the discussion window is brief, and we are concerned that other reviewers may not take the opportunity to assess our detailed reply. **Therefore, to facilitate your evaluation**, we prepared this concise summary of our work's contributions and how we addressed the main concerns.

**Summary of Contributions: CQ-DINO**

1. **Identified a Fundamental Problem:** We are the first, to our knowledge, to identify severe gradient dilution—caused by vast vocabulary sizes ($C$) —as a primary bottleneck in **vast vocabulary** object detection.
2. **Proposed an Effective Solution:** We developed an image-guided query selection module that prunes the label space from $C$ to a small subset $ C’ $. This reframes the optimization from balancing one positive against $C-1$ negatives to balancing one positive against $C’-1$ ***hard negatives***.
3. **Effective Category Modeling:** We introduced learnable category queries with a hierarchical tree structure to explicitly and efficiently model relationships between categories.

**Summary of Main Concerns and Our Responses**

1. **Deeper analysis of gradient dilution vs. class imbalance.** **(R BueR, R YWme)**

    We provided a detailed theoretical analysis clarifying that these issues have distinct origins. Our formula, $\rho_{c^+} \propto \frac{n_{c^+}\cdot \epsilon^+}{(N-n_{c^+})\cdot \epsilon^-}$, shows that class imbalance affects the numerator $n_{c^+}$, diluting gradients for rare classes. The vast vocabulary size (C) dramatically inflates the denominator $N-n_{c^+}$, diluting gradients for **both rare and common classes**. Our work is the first to isolate and address the latter.

2. **Comparison to Grounding DINO (R-BueR, R-YWme):**

    We clarified that Grounding DINO is not scalable for this task, requiring over 331 inference passes per image (768 GPU hours for the V3Det val set). To provide a meaningful comparison, we benchmarked it on V3Det's rare classes (Table R-2), where it underperforms DINO by 13.0 AP. The results reveal its training strategy fails to learn distinctions between semantically similar categories—a core challenge in vast vocabulary settings.

3. **Discussion on Query-Based Methods (R-BueR):**

    We have added a dedicated discussion of related Category Query-based Methods. We further clarified how our work is different, as our query selection design is directly motivated by our novel analysis of the gradient dilution problem during *training*, not just inference.

4. **Ablation on Training vs. Inference Contributions (R-BueR, R-YWme):**

    We conducted a detailed ablation study (Table R-1) that builds upon the Table 4 in our original paper. The results conclusively show that mitigating gradient dilution during training is the primary driver of our performance gains.

5. **Visualization of Gradient Changes (R-8AUD, R-9iMy):**

    We provided new visualizations of gradient magnitudes over extended training iterations (Table R-6, R-8). These plots empirically validate our theoretical analysis, clearly demonstrating that our method effectively rebalances gradients for both positive and hard negative samples.

6. **Analysis of Performance Bottlenecks (R-YWme):**

    We performed a detailed analysis of failure cases (Table R-4, R-5). Our findings indicate that performance on rare categories and small objects remains the primary bottleneck, which presents clear avenues for future work.

7. **Algorithmic Formulation (R-9iMy):**

     We have added formal algorithmic descriptions (Algorithms 1, 2, and 3) to enhance the clarity and reproducibility of our method.


We believe our detailed responses and new results address the reviewers' concerns well and substantially strengthen the paper. We introduce a novel perspective on the fundamental gradient dilution problem in vast vocabulary object detection. We develop CQ-DINO, a well-motivated and efficient solution. We believe our work is worth communicating and respectfully hope you will consider our work favorably.

Sincerely,

The Authors

---

### Decision · Program_Chairs · 2025-09-17

**Decision:**

Accept (poster)

**Comment:**

This paper proposes a method to mitigate gradient dilution problem in vast vocabulary object detection, where the positive categories and hard negative categories receive little learning signal compared to the overwhelming easy negatives. Reviewers appreciate the good empirical performance (BueR, YWme, 8AUD, 9iMy), well-motivated intutive idea (YWme, 8AUD, 9iMy), analysis of gradient dilution (YWme), and clear writing (BueR, 8AUD). The authors successfully addressed the concerns of all reviewers (e.g. analysis of gradient dilution, ablation on training vs inference contribution, comparison with Grounding DINO). After considering the reviews and rebuttals, AC recommends acceptance of the paper due to its clear insight, good analysis, and strong empirical results.